# Structure and function relationship of OqxB efflux pump from *Klebsiella pneumoniae*

Nagakumar Bharatham [1,2], Purnendu Bhowmik [1,2], Maho Aoki [3], Ui Okada [3], Sreevalli Sharma [1,2], Eiki Yamashita [4], Anirudh P. Shanbhag [1], Sreenath Rajagopal[1], Teby Thomas [5], Maitrayee Sarma[1], Riya Narjari[1], Savitha Nagaraj [6], Vasanthi Ramachandran [1,2], Nainesh Katagihallimath[1,2], Santanu Datta [1] & Satoshi Murakami [3]✉

OqxB is an RND (Resistance-Nodulation-Division) efflux pump that has emerged as a factor contributing to the antibiotic resistance in *Klebsiella pneumoniae*. OqxB underwent horizontal gene transfer and is now seen in other Gram-negative bacterial pathogens including *Escherichia coli*, *Enterobacter cloacae* and *Salmonella spp.*, further disseminating multi-drug resistance. In this study, we describe crystal structure of OqxB with n-dodecyl-β-D-maltoside (DDM) molecules bound in its substrate-binding pocket, at 1.85 Å resolution. We utilize this structure in computational studies to predict the key amino acids contributing to the efflux of fluoroquinolones by OqxB, distinct from analogous residues in related transporters AcrB and MexB. Finally, our complementation assays with mutated OqxB and minimum inhibitory concentration (MIC) experiments with clinical isolates of *E. coli* provide further evidence that the predicted structural features are indeed involved in ciprofloxacin efflux.

[1] Bugworks Research India Pvt. Ltd., Centre for Cellular and Molecular Platforms, GKVK, Bellary Rd, Bengaluru, Karnataka, India. [2] The University of Trans-Disciplinary Health Sciences and Technology (TDU), Bengaluru, Karnataka, India. [3] Department of Life Science and Technology, Tokyo Institute of Technology, Yokohama, Japan. [4] Institute for Protein Research, Osaka University, Suita, Osaka, Japan. [5] St. John's Research Institute, Bengaluru, Karnataka, India. [6] St. John's Medical Hospital, Bengaluru, Karnataka, India. ✉email: murakami@bio.titech.ac.jp

The COVID-19 pandemic has crippled nations worldwide and has necessitated an urgent assessment of the likely microbial pandemics in the future. One such threat is the widespread rise in antimicrobial resistance (AMR) among bacterial pathogens, and estimates suggest that it could cause up to 10 million deaths a year by 2050[1,2]. The severity of this danger is evident from the current incidence of around 800,000 deaths per year due to pneumonia, which includes more than 300,000 children. Klebsiella pneumoniae is a Gram-negative opportunistic pathogen, widely encountered as the causal organism associated with a wide variety of hospital-acquired infections; including pneumonia, bloodstream infections, wound and surgical site infections, meningitis, and urinary tract infections (UTIs). In a recent study, it was found that a major sub-group of COVID-19 patients, with a mortality rate as high as 56.7%, had secondary microbial infections and the multi-drug resistant Klebsiella pneumoniae was the predominant pathogen isolated from all of them[3]. It belongs to the ESKAPE group of pathogens (Gram-positive bacteria including Enterococcus faecium and Staphylococcus aureus; and Gram-negative bacteria including Klebsiella pneumoniae, Acinetobacter baumannii, Pseudomonas aeruginosa, Enterobacter sp.), that are continuously evolving to build resistance against different types of antibiotics[4,5]. The increasing mortality due to infections caused by resistant K. pneumoniae has caused global alarm. WHO (world health organization) has declared K. pneumoniae a priority pathogen against which there is an urgent need for developing next-generation antibiotics. Antibiotics like nitrofurantoin, β-lactams, aminoglycosides, quinolones, tigecycline, and colistin[6–9], which were potent against K. pneumoniae, are now proving to be ineffective, one explanation for this could be the increased expression of efflux pumps. Most Gram-negative bacteria possess multiple families of efflux pumps that reduce the intracellular concentrations of antibiotics, leading to intrinsic or acquired resistance against multiples classes of antibiotics[10–12]. Among the efflux pumps, members of the RND family are reported to be the significant contributors of AMR[13–16]. While the structural and functional aspects of the AcrAB-TolC pump from E. coli are well studied[17–23], the same is not known for its homolog OqxB from K. pneumoniae.

OqxB is a member of the RND efflux pump family that has been shown to confer resistance against various antibiotics, like quinolones, nitrofurantoin[5,24–27], quinoxalines, tigecycline, and chloramphenicol, detergents, and disinfectants. Besides being on the bacterial chromosome, it is also present on mobile insertion elements like IS26 or plasmids in E. coli and E. cloacae showing its potential to spread to other bacteria[27]. Though there are reports of oqxB occurrence and the resistance it confers,[24,26–28] there is little else known. The goal of this study is to determine the crystal structure of OqxB and to provide better insights into its role in antibiotic efflux.

Protein crystal structures play a definitive role in elucidating complex biological mechanisms involving substrate/inhibitor binding. They offer atomic insights related to its biological activity, thereby facilitating drug development to manipulate protein function[29]. The efflux transporter protein E. coli AcrB is no exception. The determination of its structure provided knowledge about substrate/inhibitor binding pockets along with probable routes of substrate entry and exit[30]. We have solved the crystal structure of E. coli AcrB, symmetric trimer with each monomer/protomer composed of a transmembrane region 50 Å thick and with a 70 Å protruding headpiece[17]. The E. coli AcrB transporter's structure facilitated understanding of the efflux mechanisms in other Gram-negative bacteria from ESKAPE family such as MexB, AcrB, and AdeB of P. aeruginosa, K. pneumoniae, and A. baumannii, respectively, due to their high degree sequence homology. Later, we and other groups elucidated that E. coli AcrB exists as an asymmetric trimer. Each domain has a different conformation corresponding to one of the three functional states (access/loose, binding/tight, and the third has extrusion/open conformations) of the transport cycle[31–33]. These structures revealed that drugs are exported by a three-step rotating mechanism in which substrates undergo a sequentially ordered binding before being subjected to efflux into the extracellular milieu. These asymmetric trimer structures also facilitated the modeled arrangement of AcrA and TolC and their interaction with AcrB[34,35]. The substrate-binding modes and involvement of essential amino acid residues that participate dynamically in the efflux mechanism were also identified[36–39]. Computational methodologies, like molecular docking and MD simulations methods, exploited this structural data to build the binding modes of antibiotics and decipher their export mechanisms[40,41]. The tMD (targeted molecular dynamics) simulations technique was able to link the drug-bound peristaltic movement and the associated conformational changes in the trimeric efflux pump[42–47]. The crystal structure and computational models were used to design several potent inhibitors which hinder the conformational changes and thereby defy drug efflux[48–50]. We have also utilized this structural information to understand binding modes and interaction patterns of novel antibacterial compounds with AcrB, which provided clues to chemical modifications and thereby mitigate the efflux liability[51]. Recent developments in cryo-electron microscopy (EM) in solving multi-subunit complex to its atomic details represents a paradigmshift in understanding the complexities of these sophisticated pumps[23].

The phylogenetic tree of various characterized efflux pumps indicates that they are clustered closer to each other in different clades. The pump OqxB, which is known for its ability to expel a broad variety of substrates, is seen to belong to a separate and structurally unexplored clade as compared to previously characterized proteins such as AcrB, MexB, TriC, and AdeB, which are known to lie in different clusters (Supplementary Fig. S1). In addition, recent observations indicate that OqxB seems to play a decisive role in other Gram-negative bacteria like E. coli, Enterobacter cloacae and Salmonella spp. due to its mobilization through horizontal gene transfer[26]. This qualifies it as a prime candidate for a detailed structural-functional analysis and studies its role in efflux.

Here, we report the crystal structure of the K. pneumoniae OqxB efflux pump with DDM molecules bound in its substrate-binding pocket. We have further utilized this structure to identify the key amino acids contributing to the efflux of fluoroquinolones. We further assess its efflux propensity through systematic gene complementation and biochemical evaluations. We believe that this work will lead to a better understanding of the contribution of OqxB transporters and provide clues to address the threat posed by OqxB dependent efflux liability.

## Results

**Crystal structure of the OqxB.** OqxB from K. pneumoniae was purified and its crystal structure was solved at 1.85 Å resolution. (PDB ID: 7CZ9, Supplementary Table S1). The asymmetric unit contains six OqxB monomers/protomers and each monomer is composed of 1040 residues. The overall fold of the trimer is similar to other published HAE1 sub-family of RND class of efflux pumps such as AcrB and MexB. Each OqxB monomer contains 12 trans-membrane (TM) helices (TM1 to TM12). The large periplasmic domain of the OqxB protomer that emerges from the inner membrane is connected through two extracellular loops that link TM1 with TM2 and TM7 with TM8 (Fig. 1A). The six individual protomers (Fig. 1B) are structurally similar with main chain atoms RMSD of less than 0.8 Å (Supplementary

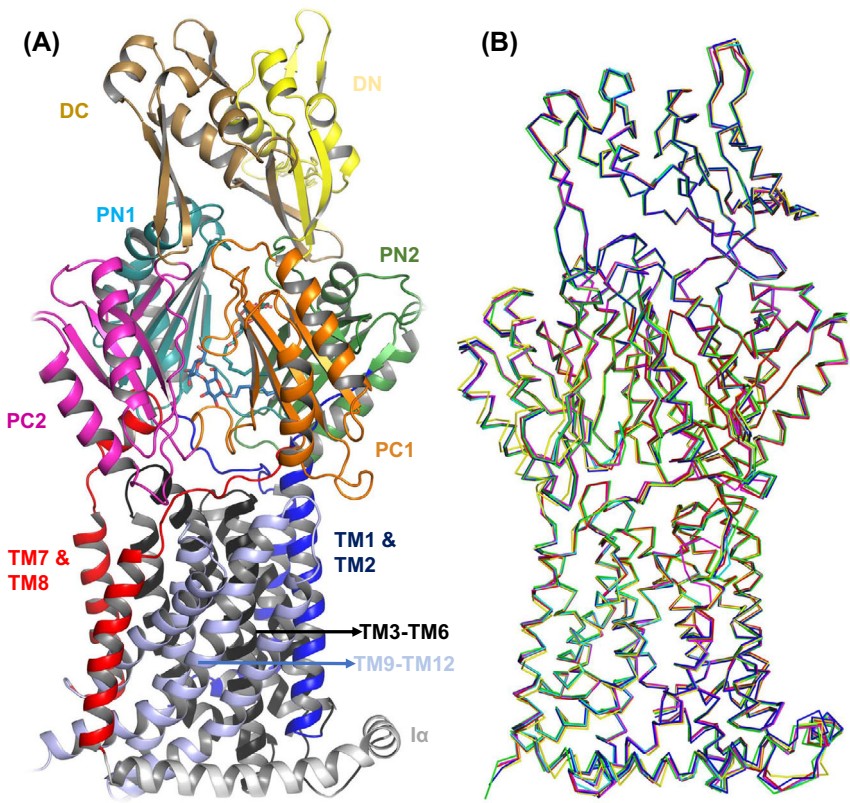

**Fig. 1 OqxB structural features.** The OqxB structure single protomer (**A**) with important sub-domains is highlighted with varied colors and labeled accordingly. The porter sub-domains PC1, PC2, PN1, and PN2 are highlighted with orange, magenta, cyan, and green color cartoon, respectively. The six OqxB protomers (**B**) superimposed to demonstrate the identical nature of individual protomers.

Table S2). Superposition of the OqxB monomer onto the AcrB asymmetric trimer (Supplementary Fig. S2) indicated that it is similar to AcrB binding/tight monomer conformation with 1.86 Å main chain atoms RMSD while it showed 2.42 Å and 2.71 Å RMSD with access/loose and extrusion/open conformations of AcrB, respectively. Similar observations were noted with all the other protomers of the OqxB structure. This analysis demonstrated that all the protomers within the two OqxB trimer molecules are highly similar and are in binding/tight conformation. The periplasmic domain or porter domain of OqxB can be divided into four sub-domains, namely PN1, PN2, PC1, and PC2. The N-terminal TM1 leads to the PN1 sub-domain that connects the PN2 sub-domain, similarly, TM7 leads to the PC1 sub-domain that connects the PC2 sub-domain. The DN and DC docking sub-domains are extensions of PN2 and PC2 sub-domains, respectively. These docking sub-domains are a pre-requisite for OqxA binding and also for the trimer formation for a functional pump. In particular, the DN docking sub-domain possesses a long loop (thumb loop) that penetrates deep into the neighboring subunit, thereby making a critical inter-protomer interaction[37]. Like other RND pumps, substrate entrance opening between PC1 and PC2 sub-domains are present in all three protomers (Fig. 2A, B). This entrance permits the substrate molecules to enter the pump via the periplasm to reach the substrate-binding pocket.

**Substrate-binding pocket comparison with other RND pumps.** The OqxB protomer substrate-binding pocket is located deep inside the periplasmic cleft, between the β-sheets of PN2 and PC1 sub-domains (Fig. 2B). Surprisingly, two DDM (n-dodecyl-β-D-maltoside) molecules are bound to each protomer of the OqxB trimer. The head group (maltoside) of the first molecule (DDM1)

occupies the hydrophilic pocket close to the exit funnel, whereas the second molecule (DDM2) head group resides near the entrance formed by PC1 and PC2 sub-domains. Irrespective of their head group position, their extended alkyl tails orient into the hydrophobic pocket formed by several phenylalanine and leucine residues (Fig. 2B, C). The head group of the DDM1 molecule forms hydrophobic interactions with the residues F180, L280, F618, L621, while its tail part interacts with the side chains of F295, F626, and F636. The tail of DDM2 forms hydrophobic interactions with the residues L138, V141, and F295. Similar observations were made with all the three OqxB protomers (Fig. 2C), and for further analyses, a single protomer was considered. The hydroxyl groups of the DDM1 terminal sugar ring form hydrogen bond interactions with two positively charged arginine residues, R48 and R157 (Fig. 2D), while the second sugar ring interacts with the three glutamine residue sidechains, i.e., Q46, Q134, and Q178.

The OqxB substrate/drug-binding pocket as compared with other known RND pumps, in particular MexB (preferred over AcrB), due to the presence of a crystal structure with DDM bound in its substrate-binding pocket (PDB ID: 3W9I). MexB crystal structures were solved with a single DDM molecule (PDB ID: 3W9I), and lauryl maltose neopentyl glycol (LMNG; PDB ID: 6IIA) independently. The orientation of a single DDM molecule bound to MexB[52] is different from that of DDM1 bound to OqxB (Supplementary Fig. S3). In MexB, the extended alkyl tail of the DDM molecule occupies the phenylalanine cage composed of the residues F136, F178, F610, F615, F617, and F628. Three out of six phenylalanine residues (F180, F618, and F636) are at identical positions of OqxB; two are replaced with leucine (L138 and L621) and one with alanine (A623). The space created due to smaller side chains (leucine and alanine) at the phenylalanine cage/

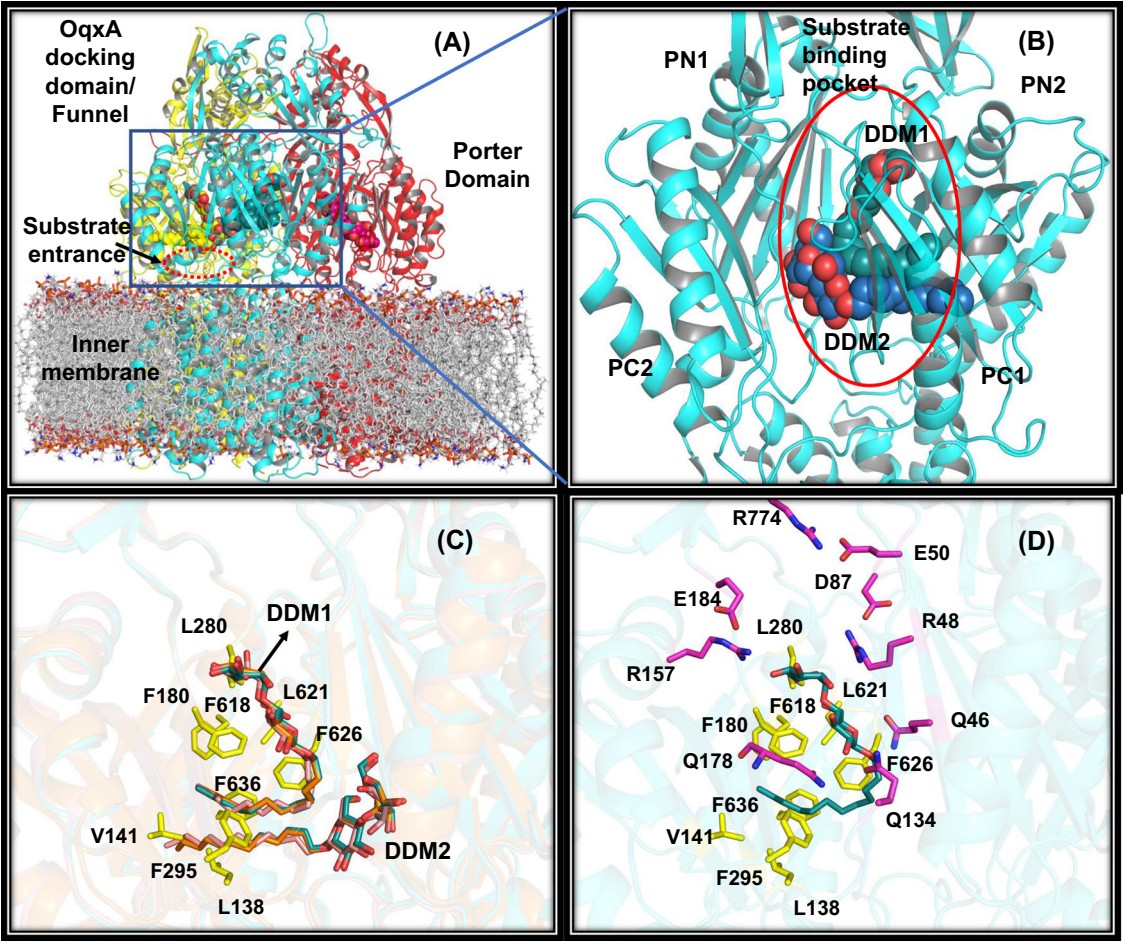

**Fig. 2 OqxB substrate-binding pocket.** The OqxB trimer is shown with a bilayer membrane model (**A**). The individual protomers are highlighted with red, cyan, and yellow color cartoons. The substrate-binding pocket with two DDM ligand molecules (**B**) is shown in one of the protomers. The DDM1 and DDM2 molecules are shown as cyan and blue colors spheres, respectively. The identical binding orientations of DDM1 and DDM2 molecules (**C**) in three individual protomers of OqxB trimer. The hydrophobic residues of the substrate-binding pocket are highlighted as yellow sticks and labeled. The hydrophilic (magenta sticks), as well as hydrophobic (yellow sticks) residues of substrate-binding pocket (**D**). DDM1 molecule shown as cyan sticks.

substrate-binding pocket was filled by two other phenylalanine residues (F295 and F626) which are unique in OqxB sequence (Supplementary Fig. S4). Several other differences are described along with Supplementary Fig. S3. These variations led to in-depth observations around the substrate/drug-binding pocket of OqxB.

The charged arginine residue, R157 present on the alpha helix of the PN2 sub-domain, is unique to OqxB and forms hydrogen bond interaction with the DDM1 molecule. R157 is replaced with a small hydrophilic serine residue (S155) in both AcrB and MexB pumps (Fig. 3). In the case of AcrB, the serine residue side chain forms water-mediated interaction with the backbone carbonyl of F178 residue that is one of the important phenylalanine cage residues. This water-mediated interaction is observed in several apo and ligand-bound AcrB crystal structures, including AcrB-peri (AcrB without transmembrane portion) structures, signifying that the water molecule is not random bulk water (Fig. 3B). Though these two residues are at identical positions in the MexB structure (Fig. 3C), the lack of water molecule densities in the structure hindered the comparison. Water molecule at the corresponding position close to R157 residue side chain in OqxB structure is unlikely due to its longer side chain (Fig. 3A). The R157 guanidino side chain forms intra-molecular interactions with the E184 residue side chain and the S182 main chain

carbonyl group, contributing to the difference in the loop and β-sheet orientation in comparison with AcrB and MexB (Fig. 3D).

Next, we compared g-loops (gate loop) that are part of the PC1 sub-domain and resides close to the entrance tunnel in OqxB, AcrB, and MexB (Fig. 4). Several structural and mutational studies demonstrate that the g-loop is important to allow the substrate molecules into the substrate-binding pocket and facilitate the efflux process[53]. In the g-loop, four glycine residues (G614, G616, G619, and G621) are present in AcrB, three glycines (G614, G616, and G621) in MexB (Supplementary Fig. S5A, B), and only one glycine, G620 (equivalent to G614 of AcrB and MexB) on OqxB loop. Interestingly, two proline residues (P619 and P630) are present on each side of the loop (Supplementary Fig. S5C) that are not observed in the other two RND family efflux pumps. Moreover, the OqxB g-loop is longer by two residues (Fig. 4A) than AcrB and MexB (16 vs. 14 g-loop residues). Two phenylalanine residues (F615 and F617) present on g-loop of both AcrB and MexB are crucial for substrate binding (Figs. 4B and 4C) were replaced with small hydrophobic residues such as L621 and A623 in OqxB. This is compensated by F626 that swings back towards substrate-binding pocket facilitated by the extended loop, as well as the presence of prolines on both sides of the loop thereby contributing to the phenylalanine cage architecture (Fig. 4D). These overall structural differences

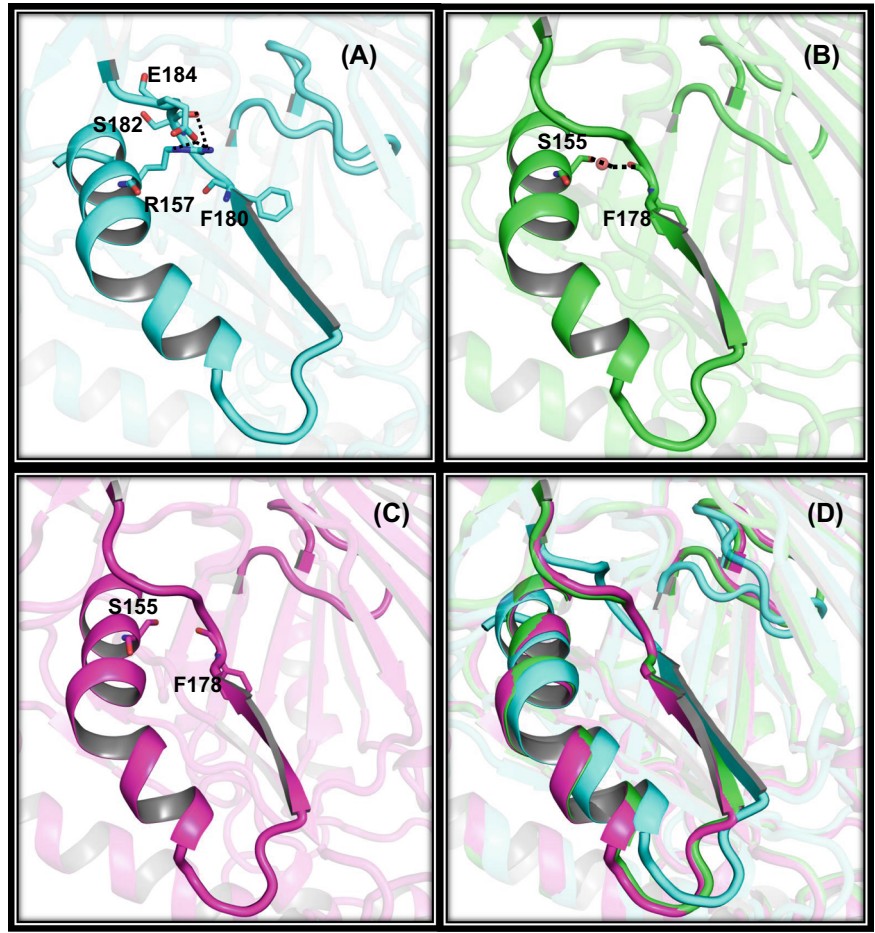

**Fig. 3 Key residue variation in OqxB substrate-binding pocket.** Lengthy positively charged arginine residue R157 and its intra-molecular interactions with E184 residue are highlighted in OqxB (**A**). The small hydrophilic residue S155 is present in AcrB (**B**) and MexB (**C**) at identical positions, superposition of all the three structures (**D**). The S155 side chain demonstrated water-mediated interaction with the F178 main chain carbonyl oxygen atom in the case of AcrB. The OqxB structure is shown with cyan cartoons whereas AcrB and MexB are shown as green and magenta cartoons, respectively. Residues labeled accordingly and intra-molecular interactions depicted as broken lines. All the three structures superimposed to highlight the conformational change at β-sheet and loop portion of OqxB. The g-loop showed in all three structures for better understanding.

are responsible for the binding mode difference of DDM molecules at OqxB substrate-binding pocket compared to MexB.

**Role of efflux pumps in fluoroquinolone resistance.** OqxB, the RND efflux pump component, is associated with increased resistance in clinical isolates *of E. coli* and *K. pneumoniae*[24]. To understand the efflux liability for fluoroquinolones in *E. coli* in the clinic, 84 recent clinical isolates were evaluated by MIC experiments (at St. John's Medical Hospital, Bengaluru, India). The MIC of ciprofloxacin against these clinical strains of *E. coli* was in the range from 0.015 to >8 µg/mL (Supplementary Table S3). Of the 84 tested, 75 were resistant to ciprofloxacin with a MIC ≥ 8 µg/mL. To delineate the cause of the resistance, the Quinolone-resistance determining regions (QRDR) of *gyrA* and *parC* were sequenced. Mutations S83L and D87N of *gyrA* and S80I or S80R, as well as E84V or E84G of *parC* that is known to confer ciprofloxacin resistance, were observed in 80 strains (Supplementary Table S3) with only four outlier strains without mutations in *gyrA* or *parC* (their MICs range from 0.015 to 0.25 µg/mL). We performed a MIC experiment to measure the contribution of efflux towards resistance by using an efflux pump inhibitor PAβN (Phenylalanine-Arginine Beta-Naphthylamide) on a panel of nine isolates (five with the above *gyrA* and *parC* mutations and four without any target mutations). None of the

strains had an altered MIC in the presence or absence of PAβN except for one susceptible clinical strain (SEC032) that showed a 4-fold reduced MIC (Supplementary Table S4). In contrast, PAβN inhibition of AcrB potentiated linezolid activity and reduced its MIC by 10 to 20-fold. This data unequivocally indicated that ciprofloxacin has minimal efflux liability through the efflux pump AcrB in these strains. We tested by PCR and target sequencing for the presence of *oqxB* amongst the 84 clinical isolates to find three isolates to possess the gene (Supplementary Table S5). The accompanying mutations in the QRDR region of these strains precluded their use in establishing the role of OqxB in the efflux of the fluoroquinolones. Therefore, complementation experiments in "sensitive strains" were undertaken to further probe the role of OqxB in efflux towards fluoroquinolone resistance.

**OqxB can form functional efflux complexes in *E. coli*.** OqxB's role was studied by heterologous expression and complementation studies in wild-type *E. coli* and its Δ*acrB* strain. Over-expression of *oqxAB* in *E. coli* BW25113 wild type (expression confirmed by quantitative Real Time-PCR, Supplementary Fig. S6) led to an 8 to 16-fold shift towards higher MIC for quinolones like ciprofloxacin, moxifloxacin, and levofloxacin (Table 1), indicating its involvement in their efflux. In addition,

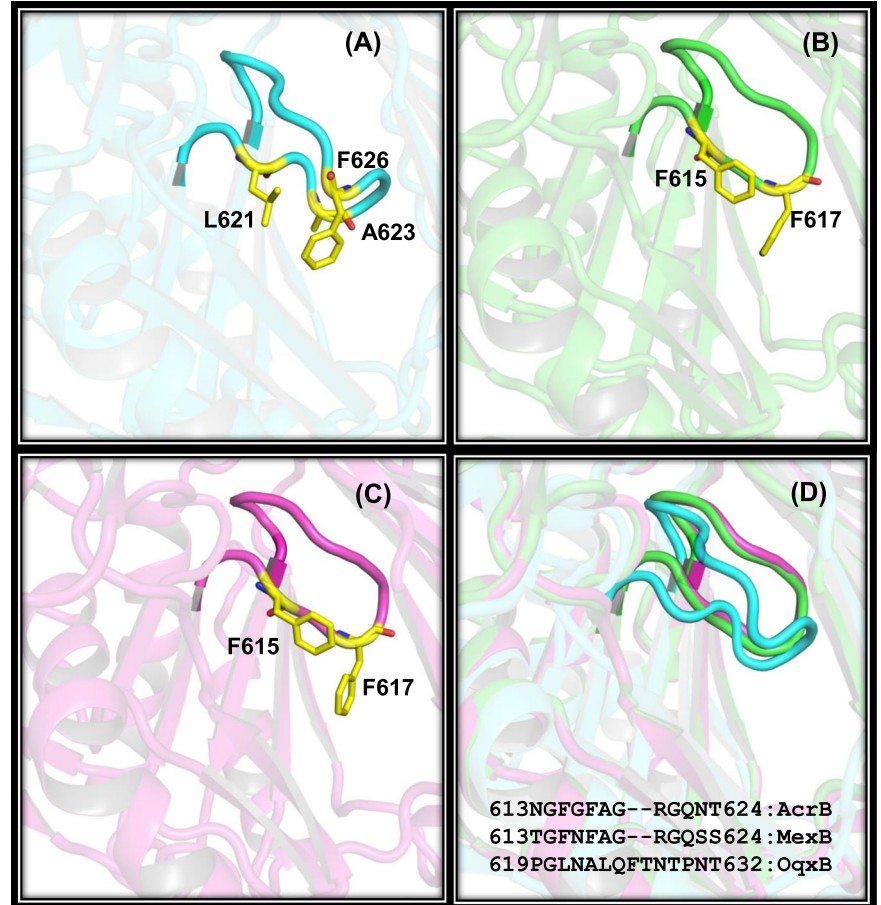

**Fig. 4 The gate loop (g-loop) variations.** Structural and sequence level differences of g-loop are shown by highlighted loop structure and important residues of OqxB (**A**), AcrB (**B**), and MexB (**C**). Important hydrophobic residues which are crucial for substrate binding were highlighted as yellow color sticks and labeled. All the three structures were superimposed to emphasize the g-loop orientational differences of OqxB comparing with the other two structures (**D**). Sequences of g-loop aligned to highlight the extra residues of OqxB.

**Table 1 MIC values with *acrB* and *oqxB* complementation in *E. coli* BW25113.**

| Compounds | *E. coli* BW25113 (WT) | *ΔacrAB* | WT + Plasmid pTrc99a | WT + *acrAB* | WT + *oqxAB* | WT + *oqxA* + *oqxB* (R157A) |
|---|---|---|---|---|---|---|
| Linezolid[#] | 400 | 12.5 | 400 | 200 | >400 | 400 |
| Novobiocin[#] | 200 | 6.25 | 400 | 200 | >400 | 200 |
| Ciprofloxacin[*] | 12.5 | 6.25 | 6.25 | 6.25 | 50 | 25 |
| Levofloxacin[*] | 25 | 6.25 | 12.5 | 12.5 | 200 | 25 |
| Moxifloxacin[*] | 25 | 12.5 | 25 | 25 | >400 | 50 |
| Rifampicin[#] | 10 | 5 | 5 | 10 | 10 | 10 |

[#]MIC values in µg/mL.
[*]MIC values in ng/mL.
*E. coli* BW25113 (WT: wildtype) or its *ΔacrAB* strain used for MIC.
pTrc99a, +*acrAB*, +*oqxAB*, +*oqxA* + *oqxB* (R157A): complementation with plasmid carrying the respective genes.
(Refer to Supplementary data for further details of the strains and constructs).

when *oqxB* was over-expressed in *E. coli* C43(DE3)-*ΔacrB*, the MIC experiments confirmed its role in the extrusion of other common antibiotics with high efflux liability like linezolid and novobiocin (Table 2). This suggests that OqxB can form a functional complex with the *E. coli* AcrA-TolC and further confirm its role in conferring antibiotic resistance against a wide spectrum of antibiotics, as reported elsewhere[54]. It can also be noted that the fluoroquinolone resistance due to efflux by AcrB is less as compared to that caused by OqxB.

**Binding mode determination of fluoroquinolones with OqxB by docking and MD simulations**. To ascertain the binding mode of OqxB with the three fluoroquinolones (ciprofloxacin, levo-floxacin, and moxifloxacin), a computational molecular docking approach followed by MD simulations was utilized. Residues around the DDM1 molecule were considered as binding pocket residues to generate the grid for docking simulations. All three fluoroquinolones arrived at a consistent binding mode at the substrate-binding pocket (Fig. 5). The fluoroquinolone carbonyl group interacts with the S182 main chain amino group (Fig. 5A, B, C) and the quinolone aromatic ring makes pi-cation interaction with the R157 guanidino side chain (Supplementary Fig. S7). A positively charged group of the fluoroquinolone molecules enter into a hydrophilic pocket near the exit funnel and forms

**Table 2 MIC values with *acrB* and *oqxB* complementation in *E. coli* C43(DE3)Δ*acrB*.**

| Compounds | *E. coli* C43(DE3) | Δ*acrB* | Δ*acrB* + pET21a | Δ*acrB* + *acrB* | Δ*acrB* + *oqxB* |
|---|---|---|---|---|---|
| Linezolid[#] | 200 | 12.5 | 12.5 | 100 | 200 |
| Novobiocin[#] | 400 | 12.5 | 12.5 | 400 | 100 |
| Ciprofloxacin[*] | 6.25 | 6.25 | 6.25 | 6.25 | 100 |
| Levofloxacin[*] | 6.25 | 12.5 | 6.25 | 12.5 | 100 |
| Moxifloxacin[*] | 6.25 | 12.5 | 12.5 | 25 | 200 |
| Rifampicin[#] | 5 | 5 | 5 | 5 | 5 |

[#]MIC values in µg/mL.
[*]MIC values in ng/mL.
*E. coli* C43(DE3) or its Δ*acrB* strain used for MIC.
pET21a, +*acrB*, or +*oqxB*: complementation with a plasmid carrying the respective genes.
(Refer to Supplementary data for further details of the strains and constructs).

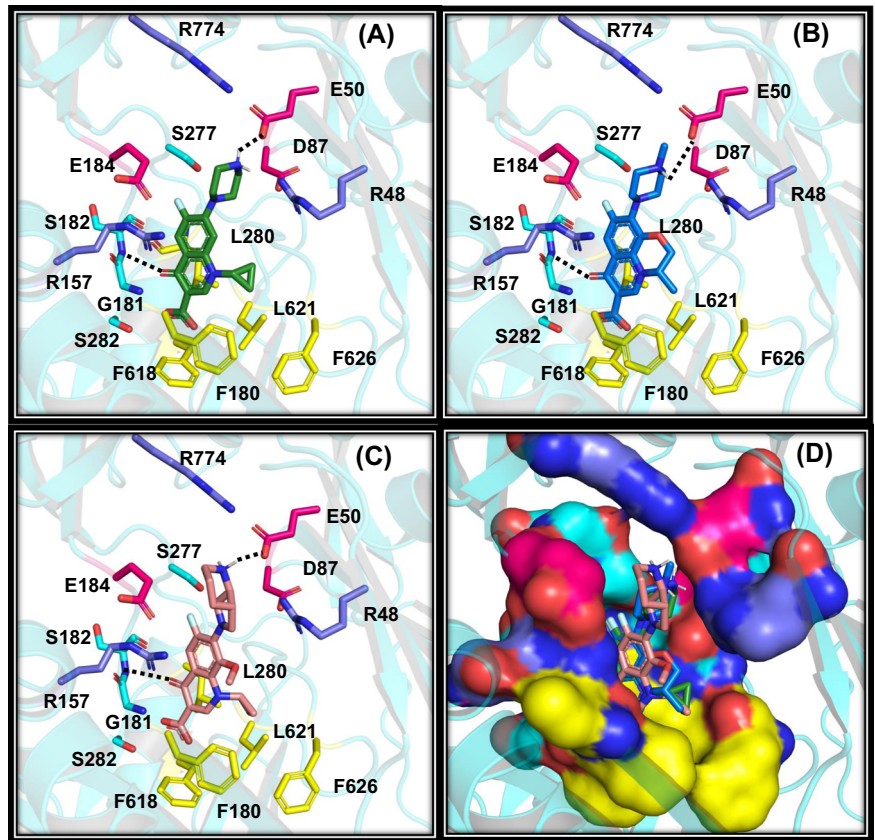

**Fig. 5 Predicted binding modes of fluoroquinolones with OqxB.** Molecular docking predicted binding orientations of ciprofloxacin (**A**), levofloxacin (**B**), and moxifloxacin (**C**). Hydrophobic, positively charged, negatively charged residues depicted as yellow, blue, and magenta sticks, respectively. Other important substrate-binding pocket residues are shown as cyan color sticks. Inter-molecular hydrogen bond interactions are highlighted with broken lines. All the predicted docking modes are superposed to illustrate similar interaction patterns of fluroquinolones with OqxB substrate-binding pocket (**D**).

hydrogen bond interaction with the E50 side chain (Fig. 5 and Supplementary Fig. S7). The OqxB hydrophilic pocket constitutes three negatively charged (E50, D87, and E184), as well as three positively charged residues (R48, R157, and R774). This highly charged OqxB hydrophilic pocket is another remarkable difference that is not observed in any other RND efflux pumps (Fig. 5 and Supplementary Fig. S7). The corresponding pocket in AcrB and MexB possess few charged residues and are majorly composed of hydrophilic residues like serine and threonine (Supplementary Fig. S4 and Fig. S8).

MD simulations initiated for membrane-embedded OqxB trimer with 6 DDM molecules were stable (Supplementary Fig. S9) with backbone RMSD (root mean square deviation) within 3 Å. Moreover, Q46 and Q134 side chains can consistently

interact with both the DDM molecules (Supplementary Fig. S10 and S11) as observed in the crystal structure. Hydrogen bond analyses of three DDM1 molecules (from all three protomers), revealed that they could form several hydrogen bond interactions with pocket residues (Supplementary Fig. S10) due to the presence of multiple hydroxyl groups on maltoside sugar rings. Though the structures were identical, variations were observed in the hydrogen bond interaction pattern of DDM molecules due to minor side-chain orientation differences, as well as its flexible nature.

To avoid variations caused by side chain differences in protomers, an alternative approach was followed to generate an initial OqxB trimer complex with a ciprofloxacin (CIPRO) molecule. OqxB trimer was constructed by superimposing a

single OqxB protomer bound to ciprofloxacin (determined by molecular docking) for MD simulations. The protein backbone RMSD of OqxB: CIPRO complex oscillates within 3 Å (Supplementary Fig. S9), while slightly elevated RMSDs were observed for CIPRO molecules compared to the molecular docking binding orientation (Supplementary Fig. S12A). Intra-molecular interactions between the important charge residues pairs (E50:R774 and R157:E184) showed stable interactions consistent with the crystal structure (Supplementary Fig. S12B). Inter-molecular interactions of CIPRO molecules with substrate-binding pocket residues within the three individual protomers (Supplementary Fig. S12C) showed 2-4 hydrogen bond interactions. All the three CIPRO molecules altered their orientations such that they interact with the R157 residue guanidino side chain. The negatively charged carboxyl and the fluoroquinolone carbonyl group of ligands formed two stable hydrogen bond interactions (>80% occupancy) with the R157 side chain (Supplementary Fig. S12C). More in-depth analyses are reported below the Supplementary Fig. S12. This evidence indicates that the CIPRO molecules interact with OqxB substrate-binding pocket, predominantly via the hydrophilic portion of the pocket, which is composed of residues, E50 and R157. To validate the proposed binding mode determined by computational methods and establish the role of key charged residues of OqxB (R157 and E50) in fluoroquinolone binding, we performed point mutation studies. The single mutant R157A demonstrated a 4 to 8-fold improvement in MIC for all the three fluoroquinolone antibiotics (Table 1) thereby signifying the importance of this residue in binding and efflux.

## Discussion

Although OqxB has a comparatively low sequence identity with other RND efflux pumps (Supplementary Table S6), our crystal structure illustrated similarities in their architecture. Several published RND efflux pump structures have revealed that they exist as an asymmetric trimer such that each protomer has a different conformation; access/loose, binding/tight, and extrusion/open. The efflux pump's asymmetric trimer explains its functional rotation mechanism for the drug/substrate extrusion process. In contrast, the OqxB structure is symmetric, wherein all the three protomers are in a binding/tight protomer state with two DDM molecules bound to each protomer. Moreover, sequence and structural comparisons provided two important pieces of evidence to speculate that OqxB can also exist as an asymmetric trimer. One of the main features of asymmetric trimer in our published AcrB crystal structure is the inclination of central α-helices[31]. This inclination triggers inter-protomer hydrogen bond interactions between conserved glutamine residues (Q108 and Q112). Visualization of several AcrB and MexB structures revealed that the conserved residues (Q104 to Q112) of the central α-helix (Supplementary Fig. S4) play a crucial role in inter-protomer interactions. While the central α-helix of OqxB is also highly conserved, similar to AcrB and MexB, such inter-protomer interactions were not discerned due to the symmetric nature of the currently determined OqxB crystal structure. A second important observation is that the triplet charged residues on the transmembrane (TM) region are conserved between OqxB, AcrB, and MexB. These triplets (two aspartates and a lysine) residues are well studied in AcrB and MexB and have been demonstrated to be crucial for proton motive force[17,55]. Aspartate residues, D410 and D411 (D407 and D408 in case of AcrB and MexB), present on TM4 helix, as well as lysine residue, K946 (K940 in AcrB and MexB) are at identical positions (Supplementary Fig. S4). Intra-molecular interactions observed between these oppositely charged residues in the OqxB structure are similar to the binding protomer of AcrB (Supplementary

Fig. S13A). Five well-ordered water molecules observed in all the six OqxB protomers interact with aspartate residues (D410 and D411) from triplets and neighboring hydrophilic, as well as charged residues such as N947, R976, E417, and R421 (Supplementary Fig. S13A). The water molecules mediated hydrogen bonding network signifies the proton path and the critical residues involved in this process. To cross-check whether such a water channel is present in other RND pumps, we have analyzed the AcrB binding protomer (PDB ID: 4DX5). Both OqxB and AcrB (Supplementary Fig. S13B) show the identical modes of water-mediated interactions with key transmembrane residues. This is a hemichannel that is typically observed in the "B/T" state of the trimer to translocate proton from the triplet to the cytosolic side.

Another exciting aspect witnessed by structural alignment studies is that OqxB DDM2 molecule occupies a probable proximal binding pocket where macrolides like erythromycin bind in RND pumps. The maltoside of DDM2 overlaps partially on the erythromycin molecule bound[56] to the AcrB access protomer (PDB ID: 3AOC), suggesting the presence of a proximal pocket in OqxB too. To probe further, erythromycin bound Neisseria gonorrhoeae (PDB ID: 6VKT) binding protomer was used for comparison (Supplementary Fig. S14A). Superposition of minocycline bound E. coli AcrB structure (PDB ID: 4DX5) clearly distinguishes distal binding pocket and proximal pockets (Supplementary Fig. S14B). Sugar rings of LMNG molecule bound to the P. aeruginosa MexB (PDB ID: 6IIA) occupy the distal binding pocket, whereas extended alkyl chains occupy the proximal binding pocket where macrocyclic molecules bind (Supplementary Fig. S14C). Similarly, the sugar rings of the OqxB DDM1 molecule reside in the distal binding pocket, and DDM2 alkyl chains occupy the proximal binding pocket of OqxB (Supplementary Fig. S14D). Most of the proximal pocket hydrophobic residues in all four structures are identical or similar (Supplementary Table S7). Our current OqxB crystal structure uncovered crucial attributes such as g-loop variations, where F626 contributes to substrate binding. Such interactions cannot be captured either from apo structures or homology modeling techniques. Though the phenylalanine cage of the substrate-binding pocket is comparatively intact, the pocket's hydrophilic portion has several charged residues showing significant variations with other RND pumps. To our knowledge, none of the RND efflux pumps demonstrate such a high charge propensity. It is also important to reiterate that intra-molecular interaction between E50 and R774 residues may play a crucial role in substrate efflux process. The presence of R157 residue and its elongated side chain's projection towards the substrate-binding pocket is also a unique feature of OqxB. Such unique structural features near the substrate-binding pocket and associated rearrangements signify the possibility of altered substrate specificity.

Several RND pumps like AcrB and MexB adopt three different monomer conformations representing the reaction cycle intermediates named access/loose (A/L), binding/tight (B/T), and extrusion/open (E/O). The recently solved Campylobacter jejuni CmeB efflux pump by x-ray crystallography has the resting state in EEE/OOO symmetric trimer[57]. Several electron microscopy (EM) structures revealed that these tripartite RND pumps withstand resting state and convert to asymmetric trimer state when pump encounters a substrate[23,58,59]. The presence or absence of other parts of the pump complex (outer membrane factor such as TolC and periplasmic membrane fusion protein like AcrA) also drives variations in resting-state symmetric trimers (access/loose and extrusion/open)[23]. OqxB symmetric trimer in BBB/TTT conformation with DDM molecules is bound at the substrate-binding pocket in this study. DDM and LMNG were reported to demonstrate an affinity for the substrate-binding pocket of

MexB[60]. We hypothesize that DDM is a probable substrate for the OqxB pump and the availability of these molecules in the crystallization conditions facilitates BBB/TTT symmetric trimer conformation. We speculate based on our cleft moment analyses by distance measurement between PC1 and PC2 sub-domain residues and initial aMD simulations data in the absence of ligands (Supplementary Table S8 and Supplementary Figs. S15–S21) that the pump will be in resting condition of EEE/ OOO symmetric conformation (more in-depth analyses are reported as Supplementary Note in the Supplementary Information file). From resting conformation E/O it will transit to B/T conformation in the presence of substrates either through transient access/loose (A/L) conformation or directly. Based on the available experimental data and conservation in the transmembrane proton transport region, OqxB can be classified as an active transport member in RND class of efflux pumps.

The fluoroquinolone class of antibiotics had higher efflux by OqxB (around 8-fold) than AcrB, as shown in the complementation experiments. The lack of impact by PAβN (efflux pump inhibitor) on the MIC for ciprofloxacin in clinical isolates of *E. coli* also accentuated the marginal efflux liability by AcrB. Previous susceptibility studies[61] using *E. coli* efflux pump knockout data also showed a moderate shift that is similar to our observations. We attributed the increased efflux to the charged residues on OqxB and propose that the zwitterionic fluoroquinolones interact with this opposite charged R157 and E50 side chains. None of the structurally well-characterized efflux pumps possess such charged residues in the substrate-binding pocket/phenylalanine cage. Point mutation of R157A reduced efflux liability for fluoroquinolone molecules as predicted, thereby validating its involvement in binding. On the same lines, a clinical isolate with an AcrB pump mutation G288D has been demonstrated to confer ciprofloxacin resistance[62]. Together, these results corroborate our view that charged residues are crucial for the optimal binding orientation of fluoroquinolone molecules in the hydrophobic cage of RND class of efflux pumps.

Overall, our OqxB crystal structure now paves avenues for an in-depth understanding of OqxB associated efflux and aid in developing specific pump inhibitors. The structural knowledge together with the understanding of ciprofloxacin interactions that we demonstrated hopefully can aid the discovery of antibiotics with reduced efflux liabilities.

## Methods

**Protein preparation**. *oqxB* of *K. pneumoniae* was amplified from genomic DNA (JCM1662) by polymerase chain reaction using KOD-plus-Neo DNA polymerase (TOYOBO, Japan) with the forward (5′-tgatgtccatatggactttttcccgcttttttatcgac-3′) and the reverse (5′-ggaattctcagtgatggtgatggtgatgggcgggcagatcctcctggac-3′) primers and inserted via NdeI and EcoRI restriction sites into a modified pET-22b (Invitrogen) expression vector. Using this method, a C-terminal hexa-histidine tag was added to aid in purification by immobilized metal affinity chromatography. DNA was sequenced to confirm the intended modifications. The resulting plasmid was transformed into *E. coli* C43(DE3) for protein expression. The transformants were grown in ten 5 L flasks at 25 °C in Davis minimal medium[63] supplemented with 0.2% glucose and 0.1% casamino acids. Expression was induced for eight hours by adding a 0.1 mM isopropyl-β-D-thiogalactopyranoside (IPTG) at an $A_{610nm}$ of 0.6. All subsequent procedures were performed at 4 °C unless indicated otherwise. Cells were harvested by centrifugation, resuspended in 50 mM Tris-HCl (pH 7.0), 0.5 mM Na-EDTA, 1 mM MgCl2, and disrupted 3 times using a Microfluidizer M-110EH (Microfluidics Corp., NM, USA) at 15,000 psi. Cell debris was removed by low-speed centrifugation at $27,000 \times g$ for 10 min. To collect membrane fractions, the supernatant was subjected to ultracentrifugation at $145,000 \times g$ for 1 h and washed with 5 mM Tris-HCl (pH 7.0), 0.5 mM EDTA. The plasma membrane was then solubilized in 50 mM Tris-HCl (pH 7.0) buffer, 10% (v/v) glycerol containing protease inhibitors (Roche), and 1% (w/v) n-dodecyl-D-maltoside (DDM, Glycon Biochemicals GmbH, Germany) on ice for 1 h. After a further step of ultracentrifugation at $145,000 \times g$ for 1 h, the detergent-solubilized fraction was harvested and incubated with Ni-sepharose at 4 °C for 1 h. The resin was washed with the buffer containing 20 mM Tris-HCl (pH 7.5), 100 mM NaCl, 25 mM imidazole, 10% (v/v) glycerol, and 0.05% (w/v) DDM. The protein was eluted from

the affinity resin with the elution buffer comprising 300 mM imidazole. Fractions containing OqxB were collected, concentrated in Amicon Stirred Cell (Merck Millipore) with 100 kDa molecular weight cut-off Omega Ultrafiltration Membrane Disc Filter (Pall Corporation, USA), and filtered with Ultrafree-MC GV Centrifugal Filter (Merck Millipore). Further purification was performed by size-exclusion chromatography (Superdex-200 Increase 10/300 GL; GE Healthcare) in the buffer containing 20 mM Tris-HCl (pH 7.5), 100 mM NaCl, 10% (v/v) glycerol, and 0.05% (w/v) DDM at the flow rate of 0.3 mL min$^{-1}$ using AKTA explorer 10S (GE Healthcare). The peak fractions were collected and concentrated in the same way as described above to ~24 mg/mL of a membrane protein for crystallization.

**Crystallization of OqxB**. OqxB crystals were grown by the sitting drop vapor diffusion technique at 25 °C. The protein solution was mixed (1:1) with a reservoir solution containing 16% polyethylene glycol 4000, 0.1 M NaCl, 100 mM MES (pH 6.3). Crystals were grown within 2–3 weeks to optimal size ($0.4 \times 0.4 \times 0.2$ mm$^3$). The concentration of glycerol was gradually increased to 30% (v/v) by soaking in several steps for optimal cryo-protection. Crystals were picked up using nylon loops (Hampton Research, CA, USA) for flash-cooling in cold nitrogen gas from a cryostat (Rigaku, Japan).

**Crystallographic data collection and structure determination**. Data sets were collected at 100 K using an EIGER hybrid photon-counting (HPC) pixel-array detector (Dectris, CH) on the BL44XU beamline at SPring-8. Diffraction images were processed with the XDS package[64,65]. Further processing was carried out with programs from the CCP4 suite[66] and Phenix[67,68]. Data collection and structure refinement statistics are summarized in Supplementary Table S1. Native data were collected at a wavelength of 0.900 Å. The crystal structure was solved by the molecular replacement method using PHASER[69] with the AcrB (4DX5) structure[70] as the search model. Model building was carried out using programs COOT[71]. Model refinement was conducted using Rosetta-Phenix[72] and Phenix[67,68]. Ramachandran analysis revealed 97.6% in the favored region and 0.26% residues in the outliers with a Molprobity[73] score of 1.70. Figures were prepared using PyMOL.

**Strains and growth conditions for complementation studies**. Bacterial strains used in this study are listed in Supplementary Table S9. The strains were grown in cation-adjusted Mueller Hinton Broth (MHB No.2 Control Cations Hi-Media) or on 1.5% Luria Bertani Agar Miller (Hi-Media) plates. Antibiotics like Amp 100 μg/ mL (Ampicillin sodium salt; Sigma-Aldrich, Catalog no.# A9518) or Kan 30 μg/mL (Kanamycin sulfate; HiMedia, Catalog no.# MB105) were used for selection wherever required. *E. coli* C43(DE3)[74] Δ*acrB* (*acrB* deletion strain) was constructed by transduction with P1 phage that was prepared using the BW25113Δ*acrB* (JW0451 from the Keio collection[75]) as a donor. Kanamycin-resistant transductants were verified to be *acrB* deletion mutants by PCR. Double Knockout Δ*acrA*-Δ*acrB* strain in *E. coli* BW25113 was generated by homologous recombination[76]. *E. coli* Top10 F' strain (ThermoFisher Scientific) was used for DNA cloning and plasmid amplification. All the cultures were grown at 37 °C under aerobic conditions.

**Plasmids and constructs for complementation studies**. The plasmids used in this study are listed in Supplementary Table S10. The genes *acrB*, *oqxB*, and polycistronic *acrAB* and *oqxAB* were PCR amplified with primers listed in Supplementary Table S11 and Table S12. *acrB*, *acrAB*, *oqxB*, and *oqxAB* were amplified from *E. coli* BW25113 and *K. pneumoniae* (JCM1662) genomic DNA, respectively, using high fidelity Phusion DNA Polymerase (ThermoFisher Scientific). The amplicons for the complementation studies for *acrB* or *oqxB* were cloned between the NdeI and XhoI restriction sites of the pET21a (Invitrogen) expression vector. While the *acrAB* and *oqxAB*, along with a constitutive promoter sequence, were fused with a vector backbone containing the pBR322_origin and Ampicillin resistance gene (from the vector pTrc99a, Genscript) using the NdeI and HindIII restriction sites. The strains used for the complementation assays were generated by transforming these plasmids as described in Table S9.

**Determination of MIC**. MIC (Minimum Inhibitory Concentration) was determined according to CLSI guidelines, as discussed elsewhere[51]. Briefly, the strains were streaked on LB-Agar plates with Ampicillin (100 μg/mL) from glycerol stocks maintained at −80 °C and allowed to grow overnight at 37 °C. A single colony of each strain was picked from the fresh plates and grown in MHB No.2 Control Cations Hi-Media (MHB) with Ampicillin (100 μg/mL). Phe-Arg β-Naphthylamide (PAβN; Sigma-Aldrich, Catalog no.# P4157) was used at a final concentration of 20 μg/mL, wherever mentioned. The AcrAB and OqxAB expression are under a constitutive promoter, and AcrB and OqxB were induced with 100 μM of IPTG (Amresco, Catalog no.# 0487-10 G). Antibiotics and their two-fold dilution series were prepared in DMSO. Three microliters compound from each of the dilutions and 150 μL ($3–7 \times 10^5$ CFU/mL) of bacterial culture were added to a flat bottom 96 well microtiter plates to obtain final antibiotic concentrations ranging from 0.03 μg/mL to 160 μg/mL. The plates were packed in gas permeable polythene bags and incubated at 37 °C for 16–18 h. MIC was determined as the concentration at which no bacterial growth was observed as ascertained by absorbance ($A_{600nm}$) with a spectrophotometer. In the case of the wild-type strain and clinical isolates,

90% growth inhibition was used for MIC determination. In the recombinant strains, the growth is compromised partially due to the presence of a plasmid and gene over-expression. Here, 80% growth inhibition results in the reduction of the test OD values to that of the media control levels, and this is used for the computation of MIC. All experiments were performed in triplicates and the data reported in this work is representative of the same.

**Site-directed mutagenesis (SDM) for OqxB R157A**. The role of polar residues like R157 in the efflux of fluoroquinolones was determined by using mutants that had the arginine replaced with non-polar Alanine (R157A). The mutation at the target R157 of *oqxB* was introduced by site-directed mutagenesis using a rolling circle PCR amplification method developed by Stratagene (Catalog #200518)[77]. In brief, p*OqxAB* plasmid carrying the *oqxAB* genes was amplified using Pfu (ThermoFisher Scientific) and the mutagenic primers described in Supplementary Table S13. The mutant strand synthesis reaction was carried out in a Thermal Cycler (Biorad) with the cycling program of 1 hold of 95 °C for 5 min, 15 cycles of 95 °C for 30 s, 55 °C for 1 min, 68 °C for 6 min and final extension of 68 °C for 10 min. The amplified product was digested with DpnI (ThermoFisher Scientific) at 37 °C for 1 h to eliminate methylated plasmid substrate. The mixture was than transformed into Top10 F' competent cells and selected on Ampicillin (100 µg/mL) containing plates. Plasmids from individual colonies were screened and the presence of mutation R157A was confirmed by sequencing. The mutated plasmid was transformed into *E. coli* BW25113 to obtain the strain BW25113 + *oqxA* + *oqxB*(R157A) and used for complementation assays.

**RNA isolation and quantitative real-time-PCR (qPCR)**. Total RNA was extracted from mid-log phase cultures by a phenol extraction method[78]. Subsequent to DNase1 treatment and extraction, the RNA was ensured to be free of DNA, protein, and solvent contamination by spectrophotometric analysis. This total RNA was used for cDNA synthesis and quantification of the target genes *acrB* and *oqxB* using gene-specific primers listed in Supplementary Table S14.

The synthesis of the cDNA was performed following the protocol from the iTaq Universal SYBR Green One-Step Kit (Bio-rad Catalog #172-5151). The quantitative-PCR (qPCR) amplification was done using an Applied Biosystems Thermocycler and 384-well optical plates. The qPCR was performed in triplicates for each of the samples using the following conditions: an initial holding temperature of 50 °C for 10 min for cDNA synthesis and the second hold of 95 °C for 30 s for the initial denaturation, followed by 40 cycles of 15 s at 95 °C and 60 °C for 1 min. The cycling was then terminated with an additional extension of 60 °C for 30 s. Fluorescence detection was performed at the annealing phase, and the dissociation curves were analyzed to confirm the amplification of a single product. The Ct values for the Threshold cycles were determined using QuantStudio™ Design and Analysis Software v1.5.1. All real-time PCR quantifications were performed simultaneously for the target genes (*acrB, oqxB*), housekeeping gene *dnaK,* and no-template controls (NTC).

**Ligand and protein preparation for docking simulations**. The 2D structures of ciprofloxacin (CIPRO), levofloxacin (LEVO), and moxifloxacin (MOXI) were prepared using the Ligprep[79] module in the Schrodinger suite. Ligprep accepts molecules in 2D format and converts them to 3D. The Epik sub-module of Ligprep was used to generate tautomers and possible ionization states for each molecule. Each ligand was manually inspected to ensure that the correct tautomer state and ionization state at physiologically relevant pH (7.4). Output structures from Ligprep were considered as input for multiple conformations generation using the MacroModel-MTLM (mixed torsional/low-mode) method. MacroModel-MTLM combines a Monte Carlo method of exploring torsional space (that efficiently locates widely separated minima on a potential energy surface) with a low-mode conformational search method that searches along energetically "soft" degrees of freedom. The OPLS_2005 force field was used, and energy minimization was performed for 500 steps using the TNCG method. The energy window for acceptable structures was set to 21 kJ/mol (default value). Conformations of the same molecule within 0.5 Å RMSD were culled. A maximum of 500 steps was allowed for Monte Carlo sampling, and a maximum of 50 steps was allowed for low mode searching. A maximum of 20 conformers per molecule was accepted.

**Molecular docking**. The Glide v8.5[80] molecular docking module (Schrodinger, 2019) in SP-mode was used to generate unbiased binding modes for fluoroquinolone derivatives. One chain out of three chains was considered for docking as all the chains are highly similar. The DDM molecule that is in proximity to residues R157 and F180 was considered to assign the substrate-binding pocket. The protein with DDM1 molecule was processed initially using the Protein Preparation Wizard in the Maestro suite from Schrodinger to add the correct protonation state and bond orders to hetero atoms of protein residues, water molecules, and the bound ligands. This processed complex was used to generate the pre-computed docking grid. The docking protocol started with the ligand's systematic conformational expansion, followed by placement in the receptor site. Minimization of the ligand in the receptor's field was then carried out using the OPLS-AA force field with the default distance-dependent dielectric. The lowest energy poses were then subjected to a Monte Carlo procedure that sampled nearby torsional minima. Poses were ranked using GlideScore, a modified version of the ChemScore function that

includes terms for steric clashes and buried polar groups. The default Van der Waal's scaling was used (1.0 for the receptor and 0.8 for the ligand). The resulted dock poses were visualized manually in PyMOL software and considered further for MD simulations.

**Molecular dynamics (MD) simulations**. Two 100 ns conventional molecular dynamics (cMD) simulations were performed with OqxB homotrimer in the presence of POPE bilayer. In the first instance, the OqxB crystal structure with six DDM molecules was considered for simulations. In a second simulation, molecular docking predicted ciprofloxacin bound structure was considered. This ciprofloxacin bound structure (monomer) was superimposed on individual chains of OqxB trimer to generate OqxB trimer with three ciprofloxacin molecules (one in each monomer). These bound structures were utilized to predict the spatial arrangements of membrane proteins with respect to the lipid bilayer's hydrocarbon core. The PPM web server[81] by the University of Michigan was employed to predict membrane insertion arrangements. The output of PPM server calculations was considered to feed the input for the CHARMM-GUI server[82]. Membrane Builder helps the user generate a series of CHARMM inputs necessary to build a protein/membrane complex for molecular dynamics simulations. The server is also capable of reading small molecule inhibitors/substrates by utilizing CHARMM General Force Field (CGFF) or Antechamber facilitates input generation for protein-bilayer complexes and protein-ligand complexes in presences of bilayer membranes. Terminal residues were patched with ACE for N-terminal whereas CT2 groups for C-terminal residue. The OqxB trimer ligand complexes were inserted into POPE (1-palmitoyl-2-oleoyl-sn-glycero-3-phosphoethanolamine), the bilayer of 242 lipid molecules, and solvated in 0.15 M KCl solution with around 42,550 water molecules. The total number of atoms in the total simulation system is around 210,000. The CHARMM36 All-atom Force Field and Amber 18 MD package were utilized to perform the simulations[83]. The membrane inserted OqxB-ligand Systems was thoroughly minimized with 10000 steps, followed by six levels of equilibration simulations (first three equilibration steps for 125 ps each, and next three equilibration steps for 500 ps each) for the complex. After structural minimization, the system was equilibrated by linearly increasing the temperature from 0 to 303.0 K using the NVT ensemble. The NPT ensemble was applied for remaining equilibration simulations with gradually lower position-harmonic restraints for the proteins. A harmonic restraint weight of 10.0 kcal/(mol·Å²) and 2.5 kcal/(mol·Å²) was applied to all protein and lipid heavy atoms, respectively, in the minimization and initial equilibration step, and gradually reduced in the following equilibration steps. The last step of equilibration was performed without any restrains. Timestep of 1 fs was used in the first 3 equilibration steps whereas the 2 fs time step was used for the remaining three equilibration steps. The production MD simulations of 100 ns were carried out unrestrained at a temperature of 303.0 K and a pressure of 1 atm with a 2 fs time step. A 12.0 Å cut-off was used as a non-bonded cut-off and bond lengths involving bonds to hydrogen atoms were constrained using the SHAKE algorithm[84]. Finally, 100 ns accelerated molecular dynamics (aMD) simulations were performed in the absence of ligands (unliganded simulations). All the above-mentioned protocol is similar for unliganded simulations except 2 ns production MD simulation performed to calculate the aMD parameters from average dihedral energy (51695.11 kcal/mol) and average total potential energy (−601088.86 kcal/mol). The total number of residues in aMD simulations system is 3128 and the total number of atoms is 235547. Using these values alphaD, EthreshD, alphaP, and EthreshP were calculated as 2189.6, 62643.11, 47109.4, and −553979.46 kcal/mol, respectively. The coordinates were stored every 20 ps for each production MD run, and these snapshots were used for RMSD and hydrogen bond analysis. Ligand RMSD was measured by utilizing the Amber analyses tool PTRAJ. Ligand RMSD values were computed by aligning the protein backbone from MD snapshots and using the "no-fit" option for ligand molecules. The hydrogen bonding analysis tool incorporated in VMD software[85] was utilized to calculate hydrogen bond interaction analyses of MD trajectories. All the other MD simulations analyses were performed by Amber analyses tool PTRAJ.

**Reporting summary**. Further information on research design is available in the Nature Research Reporting Summary linked to this article.

## Data availability

The OqxB efflux pump crystal structure solved in this study have been deposited in the Protein Data Bank under accession code PDB 7CZ9. Other coordinates of reference crystal or EM structures which we used in this study are available under accession codes PDB 4DX5, PDB 3W9I, PDB 6IIA, PDB 3AOC, PDB 6VKT, PDB 4DX6, PDB 4DX7, PDB 3AOA, PDB 3AOB, PDB 3AOD, PDB 3NOC, PDB 3NOG, PDB 3W9H, PDB 4U8V, PDB 4U8Y, PDB 4U95, PDB 4U96, PDB 5JMN, PDB 5YIL, PDB 6BAJ, PDB 6SGS, PDB 1IWG, PDB 1OY6, PDB 1OY8, PDB 1OY9, PDB 1OYD, PDB 2DHH, PDB 2DR6, PDB 2DRD, PDB 2GIF, PDB 2HRT, PDB 2J8S, PDB 2V50, PDB 3W9J, PDB 6T7S, PDB 5T0O, PDB 5LQ3, PDB 6VKS, and PDB 6OWS. Datasets which we utilized in this study are uploaded into Figshare repository and provided the DOI information below. Dataset 1 [https://doi.org/10.6084/m9.figshare.15089934.v1]: Validation report for OqxB efflux pump crystal structure. Dataset 2 [https://doi.org/10.6084/m9.figshare.15090153.v1]: OqxB expression and vector information. Dataset 3 [https://doi.org/10.6084/m9.figshare.15090162.v1]: Molecular docking input and output files. Dataset 4 [https://doi.org/10.6084/m9.figshare.15090189.v1]: Accelerated

molecular dynamics (aMD) simulations snapshots. Dataset 5 [https://doi.org/10.6084/m9.figshare.15090249.v1]: Colony PCR of clinical isolates. Dataset 6 [https://doi.org/10.6084/m9.figshare.15090210.v1]: PyMOL session files. Source data are provided with this paper.

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

## Acknowledgements

Part of this project has been supported by NIH R01 (1R01AI136803-01). The content is solely the responsibility of the authors and does not necessarily represent the official views of the National Institutes of Health. Authors thank the Centre for Cellular and Molecular Platforms (C-CAMP) for Bio-Incubation facility support. S.M. and U.O. acknowledge support by JSPS KAKENHI Grant Numbers 18H02386, 18H05396, 18K06079, and 21H02412. This research was partially supported by Platform Project for Supporting Drug Discovery and Life Science Research-Basis for Supporting Innovative Drug Discovery and Life Science Research (BINDS) from AMED (JP20am0101072) and Joint Research Committee of Institute for Protein Research, Osaka University. Synchrotron radiation experiments were performed at BL44XU of SPring-8 (2018A6500, 2018B6500, 2018B6700, 2019A6500, 2019A6700, 2019B6500, 2019B6700, 2020A6500 and 2020A6700). The authors thank Dr. Ed Griffen (MedChemica Ltd., England) for proofreading the manuscript.

## Author contributions

S.M., S.D., N.K. and N.B. designed the project. M.A., U.O., E.Y. and S.M. performed protein purification, crystallography, and crystal structure optimization and analyses. P.B., A.P.S., N.K., designed and performed the complementation and MIC experiments. S.S., T.T., M.S., S.N., V.R. and S.D. designed and performed MIC experiments on clinical isolates. R.N. generated Δ*acrAB* double knockout strain. P.B. and A.P.S. performed SDM experiments. P.B. performed the experiments on total RNA extraction and qRT-PCR. N.B. designed and performed all the computational docking and MD simulation studies. A.P.S., P.B., S.R. and N.B. performed sequence alignment, computational simulations data analyses. N.B., N.K., S.M., P.B. and S.D. wrote the manuscript. All authors revised and contributed to the manuscript.

## Competing interests

The authors declare no competing interests.
