## [Peer Review File · Nature Communications]

REVIEWER COMMENTS

Reviewer #1 (Remarks to the Author):

In this work, Bhowmik et al. presented the X-ray high-resolution structure of the OqxB RND transporter from *Klebsiella pneumoniae*, resolved as a symmetric trimer. The work provides the first structural information on this protein, which constitutes a noteworthy result. The structural information is complemented by biochemical and computational investigations aimed at defining part of the substrates' profile of this transporter. The data reported are per se highly significant, as novel structural information on a putative pharmaceutically relevant protein is delivered. However, there are several issues with the manuscript in the present form. Below, I provide a list of points the authors should consider in order to improve the quality and the impact of their work.

1. Text is quite heavy, which does not facilitate reading. Besides, English is poor. I strongly suggest the authors revise the whole manuscript making an effort at improving readability. Note that adding line and page numbers would be of help!
2. While I have no significant objections regarding the quality of the structural part, some claims about the relevance of the OqxB system for bacterial resistance to fluoroquinolones are in clear contrast with previous findings. The interpretation of the data, and the conclusions the authors derive thereof, need substantial revision. In particular:
 - a. MICs data (section "OqxB can form functional Efflux complexes in *E. coli*") are described and discussed too poorly, virtually without any reference to previous literature (only reference 72 is cited!). In particular, the conclusion that AcrB does not contribute to efflux of FQs seems to be incorrect and is contradictory to studies from many labs (those headed by Helen Zgurskaya, Laura Piddock, and Jean-Marie Pagès, to cite a few). Have the authors cross-checked if the delta *acrB* strain overproduces *acrEF*? Besides, these general conclusions are based on data on three FQ compounds. Finally, in clinical isolates, the overexpression of AcrAB is well documented. For these reasons, the authors should make an effort at framing and critically assessing their findings on the interaction of antibiotics with OqxB in the context of previous literature on the subject.
 - b. In addition to the point about MICs data, I found not really punchy the comparison of the OqxB structure with those of AcrB or MexB transporters. In particular, while a lot of details are given regarding the intra-molecular interactions between residues of the protein and inter-molecular interactions with ligands, the comparison to other transporters comes with no true clues about the link between structure and specificity. I think that making an effort in this direction could significantly increase the impact of the work. For instance, it is possible to link (some) molecular descriptors of the antibiotics used in MIC assays to explain the different susceptibilities of AcrB and OqxB in view of the differences in their structures highlighted by the authors (particularly at the putative binding sites)?

Minor points:

- tMD stands for "targeted molecular dynamics", not for "targeting" :)
- It would be useful to report in a Table the overall degrees of identity and similarity between OqxB and some main RND transporters (at least AcrB from *E. coli* and MexB from *P. aeruginosa*). I would also report these data for the distal pocket region. Finally, in Figure S4 it would be helpful to highlight all of the conserved regions using standard sequence comparison methods (even if superposed to authors' highlights of hydrophobic and hydrophilic residues of the pocket).
- Some sentences are too straight and deliver the wrong message. To give an example, in the Introduction the authors state: "[...] due to the development of antimicrobial resistance (AMR), a phenomenon that could be attributed to the activity of efflux pumps". Given the general readership of Nature Communications, sentences like this one should be revised in order to clarify that efflux systems contribute to AMR together with several additional mechanisms/systems.
- References. Some references should be updated, and some seem to have been put in the wrong context according to my knowledge of the subject:
 1. Among the interesting studies using tMD, please consider adding two recent ones by Wang and

coworkers and by Ruggerone and coworkers: see <https://doi.org/10.1021/acs.jpcc.5b11942> and <https://doi.org/10.1016/j.bbagen.2018.01.010>.

2. Ref. 43 is quite dated, also considering the increasing number of research groups working on this subject in the last decade. My suggestion is to update the bibliography by inserting some more recent reviews such as the following ones: <https://doi.org/10.1016/j.resmic.2017.12.001>; <https://doi.org/10.4155/fmc.15.173>.

3. Ref. 44 is unrelated to the role of the G-loop (section "Substrate binding pocket comparison with other RND pumps"). See e.g. doi:10.1128/AAC.02733-13.

4. In Methods, it is unclear how long was each of the six steps performed to equilibrate the complex. Please remove qualitative attributes such as "intense" and provide the details to allow the users to use your protocol.

5. In Fig. S11 it would be desirable to make explicit the comparison between OqxB and AcrB.

6. A supplementary figure would help in appreciating the overlap between the binding sites of DDM2 in OqxB and of erythromycin in AcrB.

Reviewer #2 (Remarks to the Author):

OqxAB is a very interesting RND-type efflux pump originating from *K. pneumoniae*. The *oqxAB* genes have been found in *E. coli* in clinically relevant strains (esp. UTIs) as well, mostly on plasmids, and these genes are responsible for resistance against fluoroquinolones and nitrofurantoin. The latter fact could maybe be included in the introduction, and also the reason why it is called *oqx* (Olaquinox), which has been described by Hansen, 2005 (PMID: 16359198) first (The article cited is Hansen, 2007, which was not the first description). The manuscript describes the structure and function of OqxB. The functional analysis has been published before for many of the fluoroquinolones and it confirms earlier reports that *oqxAB* is responsible for elevated resistance against fluoroquinolones and nitrofurantoin in *E. coli*. The fact that the genes are on a mobile genetic element is, especially for UTIs, clinically relevant and makes this pump a very interesting target for inhibitor development.

The manuscript describes a very high-resolution structure of the membrane protein OqxB, which was crystallized in a symmetric form. The protomers are all found in the binding state, an observation which is interesting in the light of the single-particle Cryo-EM structures solved in presence of an efflux pump inhibitor. The BBB trimeric setup might be physiologically relevant as it describes the pump in a transport active state. The authors also conducted MD and docking analysis, especially for the main substrates of this pump, the fluoroquinolones.

The manuscript describes a novel structure of this clinically relevant OqxB RND pump. It is a very interesting result. I have following questions/suggestions for the authors:

1. The structure was solved in P1 space group, yet the protomers all adopt the binding state. This is quite surprising, including the resolution obtained (1.85 Å !), and the rmsd table (table S6) shows that there are minor changes responsible for the break in three-fold (and two-fold) symmetry. Some of the figures show that side chains are sometimes different between the protomer structures, but I would like to ask the authors if there are differences which affect e.g. slight changes in subdomain conformations? It appears e.g. that protomers A and E are more alike and also E and F, compared to the other protomers.

2. For the activities of the heterologously expressed *acrB* and *oqx* genes, a Western Blot analysis would be informative to see the differences (if any) in expression levels.

3. A rmsd comparison between OqxB on the subdomain (PN, PC, DN, DC) level compared to e.g. AcrB and/or MexB will help to quantify the structural differences between these efflux pump.

4. A visual overlap of the distal binding pocket of AcrB and OqxB will be helpful, with and without a docked fluoroquinolone, to see whether binding determinants are relevant for the differences seen in the MIC measurements.

5. I suggest removing all main chain atoms (unless involved in H-bonding) to clarify the main and supplementary figures (below more specific).

6. It would be extremely helpful, and in a way increase the relevance of the docking studies, if

mutagenesis data would be implemented. E.g. substitution of the emphasized R157 and E50 would be almost a must.

7. Some discussion on the observed BBB symmetric conformation in context of the functional rotation mechanism would be helpful to put the structure into a functional context of this interesting hypothesis.

Minor remarks.

7. Title: "Role of Oqx_B efflux pump in antibiotic resistance: A structural and functional interconnect": Apart that the manuscript does not describe the role of the Oqx_B efflux pump, the second part of the title appears to be a bit artificial (structure/function relationship is often used as a phrase). It is a structure of the pump and functional determinants are discussed on basis of the structure. I would suggest changing the title.

8. Fig. 2A: There is a line going through the figure letter "(A)".

9. Fig 2A: "exit tunnel" is easily to be mistaken by the "exit gate" found in the extrusion protomers in RND pumps. I suggest changing this ("funnel" is often used).

10. Fig 2B: "Substrate binding pocket" has different letter sizes.

11. Fig 2C and D: The cartoon in the background makes the identification of the side chains very difficult (and also the main chain atoms). Consider making the cartoon more transparent or maybe even remove and only show the side chains.

12. Fig 3 shows only a bit of the differences between the Oqx_B substrate binding pocket vs. that from AcrB/MexB. The title of the figure says "Key residue variation in Oqx_B substrate pocket" (Suggestion: "substrate binding pocket" instead of substrate pocket, as is used in the main text as well), but there only a few residues shown. It is not so clear what the differences are in total between the substrate binding pockets of Oqx_B/AcrB/MexB (see also suggestion above).

13. Fig S3C and D: The legends are missing (only A and B have legends)

14. Line 49: 300,00 -> 300,000?

15. Line 93: "E. coli AcrB exists as heterotrimer", AcrB is a homotrimer at all times, but it exists as homoconformer as well as heteroconformer (but remains a homotrimer).

16. Fig S1 or maybe a separate table: % identical residues between Oqx_B and AcrB, MexB (see line 522: "Although Oqx_B has a comparatively low sequence identity with other RND efflux pumps...", should be substantiated.

17. line 187: as search model..

18. line 227/228: "Phe-Arg β-naphthylamide (PAβN) was used, wherever mentioned, at a final concentration of 20µg/ml" -> never used/mentioned in the main text/figures/tabels nor in the suppl. Materials, am I correct?

19. Line 240: "Ligand preparation for and protein preparation docking simulations:" does not sound correct...

20. Line 343: "In particular, DN domain"-> the DN subdomain

21. Line 389: "DDM1 molecule had varied orientation and did not have such deep penetration of Oqx_B...": ->DDM1 penetrates much more than DDM2? Should this be "DDM2"? And it is not so clear what is meant with "varied orientation".

22. Lines 545-548 and Fig S11: "Five well-ordered water molecules observed in all the six Oqx_B protomers which are interacting with aspartate residues (D410 and D411) from triplets and neighbouring hydrophilic as well as charge residues such as N947, R976, E417 and R421 (Supplementary Fig. S11)". I am wondering if accessibility for protons from the periplasm via water channels in the B conformation is also visible? According what I understood from the previous literature, it is suggested that protons enter the B conformation and protonate the Asp side chains, which leads to a change to the extrusion protomer?

23. I suggest the authors look for the consequent use of capital/small letters in the manuscript e.g. Efflux/efflux, G-loop/g-loop, Glycine/glycine (Fig. S5), Wild Type/wild type, but there are more. These appear in the middle of the sentence.

Reviewer #3 (Remarks to the Author):

This paper details the structure of the RND drug efflux protein OqxB from *Klebsiella pneumoniae*. Other protein members from this transport family have been well characterized, in some cases by authors of this manuscript. Here they compare and contrast what has been achieved primarily against the proteins MexB from *Pseudomonas aeruginosa* and AcrB from *Escherichia coli*.

The RND family of proteins have very interesting structures. They have voluminous substrate binding pockets that can usually accommodate a large number of distinct substrates, through interactions with specific subsets of residues. Additionally, they have been found to function using a mechanism known as functionally rotating, where each of the three individual protomers of the trimeric structure are in three different poses depending on their involvement in the efflux process. These poses are known as access, binding and extrusion.

In this manuscript the structure of OqxB has been resolved through crystallography to 1.85 angstroms. Analysis of this structure revealed that it carried two detergent molecules (DDM1 and DDM2) within its proposed binding pocket. Although the basic structure of RND proteins is similar, it is only through detailed studies that the intricacies of how each protein interacts with its specific set of substrates occurs. Here there is detailed description of the positioning of the DMM molecules within the binding pocket. However, the link to the functionality of the protein has not been made, or it has then it is not clear. The authors compare OqxB bound to DDM and the MexB bound DDM structure. These structural inferences have been extended for ciprofloxacin using MD simulations. From this they make predictions about substrate binding residues. These predictions may well be correct (a structure of OqxB with a known substrate is required for this). However, the reasoning behind why they believe this is fruitful needs more explanation. Is DDM a substrate for the OqxB protein or a contaminant during crystallography? If it is a substrate then can the authors explain why the protein is still in a symmetrical formation? I understood that RND proteins are in a symmetric "resting" state in the absence of substrate. However, when bound to a ligand they then separate into the different functional protomers. Thus, it would be good if authors could explain functional or physiological relevance of a symmetric homotrimer of OqxB with all monomers in binding conformation with two bound DDM molecules in each monomer; or if the occurrence of this structure is because of non-physiological conditions of crystallization. If the latter then could the authors provide MD simulation evidence, as they have shown the same thing for AcrB, that OqxB can exist as a functionally relevant asymmetric trimer.

A number of sentences require references. For example, sentence three in the introduction and in the results when the lauryl maltose neopentyl glycol MexB structure is discussed.

Materials and Methods-

Why is an efflux inhibitor added- are results described using this? If DMSO is used as a solvent for some antimicrobial compounds what is it really also used as a diluent for the 2-fold serial dilutions? Why was the MIC considered as 80% growth? This seems to be an arbitrary value. Why not a standard method?

MD methodology -This sentence was copied directly from the website "The PPM server calculates rotational and translational positions of transmembrane and peripheral proteins in membranes using their 3D structure (PDB coordinate file) as input."

Is such detail required? "This server generates the input files required for various kinds of MD simulations, including bilayer embedded protein complex simulations."

Results

What is the AcrBperi structure?

"To comprehend the binding mode of fluoroquinolones, computational molecular docking approach was utilized. Residues around DDM1 molecule were considered as binding pocket residues and docking grid

was generated for docking simulations." Does this then mean that DDM2 is an artefact?

"The asymmetric unit consists of six OqxB monomers/protomers and each monomer composed of 1040 residues." Really? This is contrasting with the authors' explanation of the OqxB structure (for example in the first paragraph of discussion, where they say it is symmetric structure). The information they have provided indicates that it is indeed symmetric structure.

The authors show that OqxB is functional in *E. coli* with additive efflux capacity for fluoroquinolone antibiotics in the WT and partnering ability with periplasmic adaptor and outer membrane channel components of the AcrAB-TolC system. However, the explanation and microbiology used is very hard to follow and needs to be rewritten. As it currently is Tables 1 and 2 do not easily stand on their own. This is primarily due to the nomenclature of deletion strains that is vague

As an example:

C43(DE3) (Δ), C43(DE3) (Δ) *acrB*_OE, and C43(DE3) (Δ) *oqxB*_OE. Understanding of these strains is difficult with this nomenclature.

pEcoAB – which protein is the histag on?

Can the authors provide information that OqxB and OqxAB are expressed in the complemented strains for example by showing Western blot analysis pictures or stating some further information in this section?

What is the rationale for complementing the wildtype *E. coli* strain bearing a natural chromosomal mediated *acrAB*-*tolC* system with a recombinant *acrAB* allele, and importantly what is the authors justification that this complementation has resulted in reduction of resistance to some of the AcrB substrates for example LZD, Cipro and Levo?

Tables S4 and S5 need footnotes explaining the shading and underlining. Table S6 is not formatted correctly. What are the headers for the columns/rows?

Figure S3 what are panels C and D?

"This highly charged propensity of OqxB hydrophilic pocket is another remarkable difference that is not observed in any other RND efflux pumps (Supplementary Fig. S7)." Has the correct figure been referred to here?

Overall, the English needs to be corrected. The article (the or a) is missing in multiple sentences. Italics needs to be checked when talking about genes and IS elements.

Overall, the new crystal structure adds substantial information to the current knowledge on RND efflux transporter proteins and highlights important differences in the local structures in the RND type proteins that contribute to substrate specificity of different RND pumps.

While this information is important, in its current state the work does not open any new avenue in the field of efflux transporter research. Furthermore, integration of the biological and structural data is lacking in this paper.

Reviewer 1:

Text is quite heavy, which does not facilitate reading. Besides, English is poor. I strongly suggest the authors revise the whole manuscript making an effort at improving readability. Note that adding line and page numbers would be of help!

We have revised the manuscript by correcting the language to improve readability and added page numbers as suggested.

While I have no significant objections regarding the quality of the structural part, some claims about the relevance of the OqxB system for bacterial resistance to fluoroquinolones are in clear contrast with previous findings. The interpretation of the data, and the conclusions the authors derive thereof, need substantial revision. In particular:

a. MICs data (section "OqxB can form functional Efflux complexes in *E. coli*") are described and discussed too poorly, virtually without any reference to previous literature (only reference 72 is cited!). In particular, the conclusion that AcrB does not contribute to efflux of FQs seems to be incorrect and is contradictory to studies from many labs (those headed by Helen Zgurskaya, Laura Piddock, and Jean-Marie Pagès, to cite a few). Have the authors cross-checked if the delta *acrB* strain overproduces *acrEF*? Besides, these general conclusions are based on data on three FQ compounds. Finally, in clinical isolates, the overexpression of AcrAB is well documented. For these reasons, the authors should make an effort at framing and critically assessing their findings on the interaction of antibiotics with OqxB in the context of previous literature on the subject.

We appreciate the reviewer concern, and we have performed additional experiments and updated the results and discussion. We normally consider ciprofloxacin as a negative control in antibacterial screening experiments against wild type BW and ATCC *E. coli* strains. These compounds exhibit around 2 to 4-fold efflux liability in *E. coli* and other Gram-negative pathogens of the ESKAPE spectrum. We generally consider >8-fold MIC variation between WT BW strain and AcrB knockout strain (or in the presence of AcrB inhibitor) as efflux liability.

We agree with the reviewer that the possible overexpression of AcrAB in clinical strains. To check this, we have considered 84 recent *E. coli* clinical isolates from St. John's Medical Hospital, Bengaluru, India. The major issue with current clinical strains is high resistance frequency for the FQ class of antibiotics through the main target (GyrA/ParC) mutations. Fortunately, four strains are sensitive to FQ class, and we ascertained this by sequencing the main targets. MIC measurement in the presence of pan-efflux pump inhibitor, i.e., PAβN, demonstrated maximum 4-fold efflux liability, which is well correlated with earlier documented data [CLINICAL MICROBIOLOGY REVIEWS, 2006, 382–402]. This PAβN insensitivity additionally answers the issue of *acrEF* overexpression raised by the reviewer. We have now included all these observations in the revised version to strengthen the fact that FQ class molecules possess relatively high efflux through OqxB.

b. In addition to the point about MICs data, I found not really punchy the comparison of the OqxB structure with those of AcrB or MexB transporters. In particular, while a lot of details are given regarding the intra-molecular interactions between residues of the protein and inter-molecular interactions with ligands, the comparison to other transporters comes with no true clues about the link between structure and specificity. I think that making an effort in this direction could significantly increase the impact of the work. For instance, it is possible to

link (some) molecular descriptors of the antibiotics used in MIC assays to explain the different susceptibilities of AcrB and OqxB in view of the differences in their structures highlighted by the authors (particularly at the putative binding sites)?

We have discussed differences around distal binding pocket in Figure 3 and Figure 4 as these are significant variations and may have a key role in substrate binding and specificity. For example, the R157 side chain is unique in OqxB, where S155 present in AcrB and MexB. We speculated that R157 is crucial for fluoroquinolone binding. Now in revision, we have performed site-directed mutagenesis experiments (R157A) and demonstrated the reduced efflux of fluoroquinolones. We have also compared the fluoroquinolone binding residues of OqxB with AcrB to show the differences and added them as a supplementary figure (Figure S8).

Minor points:

-tMD stands for "targeted molecular dynamics", not for "targeting" :)

Typo correction made.

-It would be useful to report in a Table the overall degrees of identity and similarity between OqxB and some main RND transporters (at least AcrB from E.coli and MexB from P. aeruginosa). I would also report these data for the distal pocket region. Finally, in Figure S4 it would be helpful to highlight all of the conserved regions using standard sequence comparison methods (even if superposed to authors' highlights of hydrophobic and hydrophilic residues of the pocket).

We have provided the supplementary table (Table S7) with overall identity and similarity values between OqxB and other RND pumps. We have also updated Figure S4 as the reviewer suggested, which helped improve the figure's quality.

-Some sentences are too straight and deliver the wrong message. To give an example, in the Introduction the authors state: "[...] due to the development of antimicrobial resistance (AMR), a phenomenon that could be attributed to the activity of efflux pumps". Given the general readership of Nature Communications, sentences like this one should be revised in order to clarify that efflux systems contribute to AMR together with several additional mechanisms/systems.

We have corrected the sentence to improve the readability.

-References. Some references should be updated, and some seem to have been put in the wrong context according to my knowledge of the subject:

1. Among the interesting studies using tMD, please consider adding two recent ones by Wang and coworkers and by Ruggerone and coworkers: see <https://doi.org/10.1021/acs.jpcc.5b11942> and <https://doi.org/10.1016/j.bbagen.2018.01.010>.

Updated the references as suggested.

2. Ref. 43 is quite dated, also considering the increasing number of research groups working on this subject in the last decade. My suggestion is to update the bibliography by inserting

some more recent reviews such as the following ones:
<https://doi.org/10.1016/j.resmic.2017.12.001>; <https://doi.org/10.4155/fmc.15.173>.

Updated the references as suggested.

3. Ref. 44 is unrelated to the role of the G-loop (section "Substrate binding pocket comparison with other RND pumps"). See e.g. doi:10.1128/AAC.02733-13.

Agree with the reviewer and updated the reference.

4. In Methods, it is unclear how long was each of the six steps performed to equilibrate the complex. Please remove qualitative attributes such as "intense" and provide the details to allow the users to use your protocol.

Details appended in the methods section.

5. In Fig. S11 it would be desirable to make explicit the comparison between OqxB and AcrB.

We have updated the figure and compared it with AcrB (PDB ID:4DX5) binding protomer as suggested. The current Figure number is Fig. S20.

6. A supplementary figure would help in appreciating the overlap between the binding sites of DDM2 in OqxB and of erythromycin in AcrB.

We have utilized PDB:6VKT, *Neisseria gonorrhoeae* MtrD with Erythromycin bound to B/T protomer and compared with B/T protomers of OqxB, AcrB and MexB. We demonstrated that several proximal binding pocket hydrophobic residues are identical to all four RND pumps. New Supplementary table (Table S8) and figure (Fig. S21) added and cited in discussion.

Reviewer 2:

1. The structure was solved in P1 space group, yet the protomers all adopt the binding state. This is quite surprising, including the resolution obtained (1.85 Å !), and the rmsd table (table S6) shows that there are minor changes responsible for the break in three-fold (and two-fold) symmetry. Some of the figures show that side chains are sometimes different between the protomer structures, but I would like to ask the authors if there are differences which affect e.g. slight changes in subdomain conformations? It appears e.g. that protomers A and E are more alike and also E and F, compared to the other protomers.

Thank you for encouraging us to discuss this important point. Indeed, with this crystal form (P1), we can compare structural differences between all the protomers including in the asymmetric unit. In our comparison (table S2), all the protomers have the B/T conformation (RMSD=0.58±0.1). The small variation of RMSD value might be caused by the crystal packing affected from the neighbouring protomer in the crystal lattice. We expect such small variation (<1Å) is not critical and may not cause conformational variations.

2. For the activities of the heterologously expressed *acrB* and *oqx* genes, a Western Blot analysis would be informative to see the differences (if any) in expression levels.

Antibodies against AcrB and OqxB are not available commercially to perform Western-blot experiments for estimating protein expression levels. Therefore, mRNA expression was estimated by qRT-PCR in the three strains: BW25113 (wild-type) and its *acrB* and *oqx* overexpression strains. The Ct values were compared with the house-keeping gene *dnaK* (a highly expressed gene with about 38000 protein molecules per cell doi: [10.1128/AEM.69.6.3231-3237.2003](https://doi.org/10.1128/AEM.69.6.3231-3237.2003)).

The ΔCt (Ct target gene - Ct *dnaK*) and Fold Change ($2^{\Delta Ct}$) in the table below indicate that the expression of *oqx* is about 2.3-fold less than *dnaK*, whereas *acrB* in the overexpressed strain is 6.8-fold less than *dnaK*. In the wild-type *E. coli* BW25113, *acrB* is 36.6-fold less than *dnaK*. This data suggests that *acrB* and *oqx* are overexpressed in the recombinant *E. coli*.

Strain	Gene	ΔCt	Fold Change = $2^{\Delta Ct}$
BW25113	acrB	5.2	36.6
BW+AcrAB_OE	acrB	2.8	6.8
BW+OqxAB_OE	oqx	1.2	2.3

3. A rmsd comparison between OqxB on the subdomain (PN, PC, DN, DC) level compared to e.g. AcrB and/or MexB will help to quantify the structural differences between these efflux pump.

As suggested by you we have attempted to compare the subdomain by “Super” command implemented in PyMOL software. We have considered AcrB (PDB ID: 4DX5) and MexB (PDB ID: 3W9J) for comparison with OqxB. We have considered all the chains present in the PDB for the analyses (Subdomain-align-test.xls: Sheet name: OqxB_AcrB_MexB). We have considered four porter sub-domains (PN1, PN2, PC1 & PC2) as well as two docking sub-domains (DN and DC). No major variations observed in docking sub-domains. Our observations revealed that PN1 and PN2 sub-domains are similar between OqxB all chains (binding protomer) and binding protomers (highlighted with red colour) of AcrB (B-chain) as well as MexB (B and E chains). Such consistent trend not observed in case of PC1 and PC2 sub-domains. In particular, the PC1 sub-domains showed comparatively high RMSD aligning with OqxB sub-domains. The other variation is PC2 sub-domain of MexB chain-E demonstrated high RMSD comparing with any other. We suspected that the loops (such as G-loop) which are present in these domains might be causing these RMSD variations. We have omitted the loop portions in RMSD calculations. We have not observed any improvement in RMSD values for PC1 sub-domains, instead we observed better RMSD with MexB PC1 sub-domain of Access protomer (A and D chains). No improvement observed in PC2 sub-domain of MexB chain-E.

Next, we attempted aligning all the four sub-domains (PN1+PN2+PC1+PC2) together (Sheet name: OqxB_AcrB_MexB-full) where we observed that binding protomers aligning well on OqxB binding protomers. Similar observations recorded without loops where low RMSDs observed between binding protomers compared to other protomers of AcrB and MexB. These discrepancies might be due to rigid body alignment of individual domains.

To check whether the high RMSD phenomenon only related to OqxB PC1 and PC2 domains or can be observed in between other efflux pumps, we have considered two AcrB crystal structures (PDB IDs: 4DX7 and 3W9H) as well as MexB (PDB ID: 3W9J). The sub-domains of these structures were aligned on sub-domains of AcrB (PDB ID: 4DX5). We have noted the following important points from the sub-domain alignments (Sheet name: AcrB).

- Comparing binding protomer sub-domains of 4DX7 (doxorubicin bound in the substrate binding pocket) and 3W9H (pump inhibitor bound) with 4DX5 (minocycline bound) revealed that pump inhibitor bound structure shown higher RMSDs in all the four sub-domains (0.3 to 0.5Å higher). This signifies though the protein is same the RMSD variation might be caused by type of substrate/inhibitor bound and associated local structural re-arrangements.
- PN1 and PN2 sub-domains of MexB showed low RMSD with respective protomer sub-domains of AcrB, but PC1 and PC2 sub-domains showed deviations. For example, all MexB protomers PC1 sub-domains showed high RMSDs against AcrB chain A, whereas showed low RMSDs against AcrB chain C (extrusion protomer). In case of PC2 low RMSDs recorded between MexB and AcrB protomers.
- Whereas aligning all the four sub-domains (PN1+PN2+PC1+PC2) together provided proper RMSD values as expected that access protomers of MexB (chain

- A and D) showed less RMSDs 1.911 and 1.899Å, respectively against AcrB access protomer (Chain A).
- Binding protomers of MexB (chain B and E) showed less RMSDs 1.328 and 1.489Å, respectively against AcrB binding protomer (Chain B). Other protomers showed 1.5 to 2 Å higher RMSDs.
 - Similarly, extrusion protomers of MexB (chain C and F) showed less RMSDs 1.038 and 0.986Å, respectively against AcrB extrusion protomer (Chain C). Other protomers showed 0.7 to 2 Å higher RMSDs.

All these evidences signify that individual sub-domain alignment method may not be helpful to identify the structural differences.

4. A visual overlap of the distal binding pocket of AcrB and OqxB will be helpful, with and without a docked fluoroquinolone, to see whether binding determinants are relevant for the differences seen in the MIC measurements.

We have added a new supplementary figure (Figure S8) as suggested.

5. I suggest removing all main chain atoms (unless involved in H-bonding) to clarify the main and supplementary figures (below more specific).

Thanks for the great suggestion to improve the quality of the figures. We have considered the reviewer's suggestion and made necessary modifications.

6. It would be extremely helpful, and in a way increase the relevance of the docking studies, if mutagenesis data would be implemented. E.g. substitution of the emphasized R157 and E50 would be almost a must.

As suggested by the reviewer, we have generated R157A mutation and observed a 4 to 8-fold improvement in MIC for all three fluoroquinolone antibiotics (Table 1). This result signifies the importance of this residue in binding and efflux.

7. Some discussion on the observed BBB symmetric conformation in context of the functional rotation mechanism would be helpful to put the structure into a functional context of this interesting hypothesis.

We have added the relevant data in results and discussion sections based on additional MD simulations of OqxB without DDM molecules.

Minor remarks.

7. Title: "Role of OqxB efflux pump in antibiotic resistance: A structural and functional interconnect": Apart that the manuscript does not describe the role of the OqxB efflux pump, the second part of the title appears to be a bit artificial (structure/function relationship is often used as a phrase). It is a structure of the pump and functional determinants are discussed on basis of the structure. I would suggest changing the title.

Aligning with reviewer suggestion and based on our structural and mutational data, we have changed the title to:

" Structure and Function relationship of OqxB from *Klebsiella pneumoniae*".

8. Fig. 2A: There is a line going through the figure letter "(A)".

Corrected and aligned properly.

9. Fig 2A: "exit tunnel" is easily to be mistaken by the "exit gate" found in the extrusion protomers in RND pumps. I suggest changing this ("funnel" is often used).

We agree with the reviewer comment and correction made.

10. Fig 2B: "Substrate binding pocket" has different letter sizes.

Corrected and checked the font in other panels and remaining figures for consistency.

11. Fig 2C and D: The cartoon in the background makes the identification of the side chains very difficult (and also the main chain atoms). Consider making the cartoon more transparent or maybe even remove and only show the side chains.

We have removed the main chain atoms for clarity. Also made cartoon more transparent. These changes we made for all figures in the main text and supplementary figures.

12. Fig 3 shows only a bit of the differences between the OqxB substrate binding pocket vs. that from AcrB/MexB. The title of the figure says "Key residue variation in OqxB substrate pocket" (Suggestion: "substrate binding pocket" instead of substrate pocket, as is used in the main text as well), but there only a few residues shown. It is not so clear what the differences are in total between the substrate binding pockets of OqxB/AcrB/MexB (see also suggestion above).

Thank you for the suggestion; we have changed as suggested from substrate pocket to substrate-binding pocket in the main text and figure captions. Though we have shown few residues, such as R157 of OqxB and S155 in AcrB and MexB, this is one key replacement that causes rearrangements in substrate binding pocket. The presence of arginine residue and its intra-molecular interactions are responsible for the difference in the loop and β -sheet orientation compared with AcrB and MexB.

13. Fig S3C and D: The legends are missing (only A and B have legends)

Thanks for the alert; we have updated the missing legends in the revised version.

14. Line 49: 300,00 -> 300,000?

Incorporated the correction.

15. Line 93: "E. coli AcrB exists as heterotrimer", AcrB is a homotrimer at all times, but it exists as homoconformer as well as heteroconformer (but remains a homotrimer).

We agree with the reviewer...have made the necessary corrections by changing the symmetric trimer and asymmetric trimer.

16. Fig S1 or maybe a separate table: % identical residues between OqxB and AcrB, MexB (see line 522: "Although OqxB has a comparatively low sequence identity with other RND efflux pumps.... ", should be substantiated.

We have provided the supplementary table (Table S7) with overall identity and similarity values between OqxB and other RND pumps.

17. line 187: as search model..

Corrected.

18. line 227/228: "Phe-Arg β -naphthylamide (PA β N) was used, wherever mentioned, at a final concentration of 20 μ g/ml "-> never used/mentioned in the main text/figures/tables nor in the suppl. Materials, am I correct?

Thanks for your alert... in the revised manuscript we have added the clinical strains MIC data in the presence of 20 μ g/mL of Phe-Arg β -naphthylamide (PA β N).

19. Line 240: "Ligand preparation for and protein preparation docking simulations:" does not sound correct...

Sentence corrected as "Ligand and protein preparation for docking simulations".

20. Line 343: "In particular, DN domain" -> the DN subdomain

Corrected and checked for consistency throughout the manuscript.

21. Line 389: "DDM1 molecule had varied orientation and did not have such deep penetration of OqxB...": ->DDM1 penetrates much more than DDM2? Should this be "DDM2"? And it is not so clear what is meant with "varied orientation".

This analysis highlighted the DDM binding differences between OqxB and MexB. The sentence rewritten as " The OqxB DDM1 molecule does not have such deep penetration compared to MexB DDM molecule " to avoid confusion.

22. Lines 545-548 and Fig S11: "Five well-ordered water molecules observed in all the six OqxB protomers which are interacting with aspartate residues (D410 and D411) from triplets and neighbouring hydrophilic as well as charge residues such as N947, R976, E417 and R421 (Supplementary Fig. S11). ". I am wondering if accessibility for protons from the periplasm via water channels in the B conformation is also visible? According what I understood from the previous literature, it is suggested that protons enter the B conformation and protonate the Asp side chains, which leads to a change to the extrusion protomer?

To understand better and clarify whether the water channel present in the Binding protomer of other RND pumps or not, we have considered binding protomer from the

AcrB crystal structure (PDB ID: 4DX5). Comparison OqxB and AcrB (Figure S20) demonstrated that both the structures show an identical mode of water-mediated interactions with key transmembrane residues.

23. I suggest the authors look for the consequent use of capital/small letters in the manuscript e.g. Efflux/efflux, G-loop/g-loop, Glycine/glycine (Fig. S5), Wild Type/wild type, but there are more. These appear in the middle of the sentence.

Thanks for the alert... we have checked and corrected these typo errors.

Reviewer 3:

Here there is detailed description of the positioning of the DDM molecules within the binding pocket. However, the link to the functionality of the protein has not been made, or it has then it is not clear. The authors compare OqxB bound to DDM and the MexB bound DDM structure. These structural inferences have been extended for ciprofloxacin using MD simulations. From this they make predictions about substrate binding residues. These predictions may well be correct (a structure of OqxB with a known substrate is required for this). However, the reasoning behind why they believe this is fruitful needs more explanation. Is DDM a substrate for the OqxB protein or a contaminant during crystallography?

The referee raised a valid point...we believe that though DDM is an essential buffer additive, DDM and its analogues like LMNG (lauryl maltose neopentyl glycol) might be substrates for some of the RND efflux pumps like MexB (Crystal structures of multidrug efflux pump MexB bound with high-molecular-mass compounds, 2019 Sci Rep 9: 4359-4359). Though the DDM bound OqxB structure might have weak functional relevance, the bound DDM molecules provided valuable insights into the substrate-binding pocket, and residual variations with other well studied RND efflux pumps. As mentioned in the discussion, side-chain orientations of some of the variant residues, i.e., R157 and F626 and their role in substrate binding, cannot be understood with homology models Apo structures. We have attempted to determine the binding mode of Olaquinox (a well-known OqxB substrate molecule) with the current OqxB structure by molecular docking method.

Analyses of binding mode revealed the Olaquinox show a similar kind of interaction pattern as Ciprofloxacin. The negatively charged 4-Oxido group formed charge-charge or hydrogen bond interaction with the R157 side chain, whereas the hydroxyl group predicted to interact with R48 and D87 side chains. These observations were consistent with our claims that charged residues, which are the unique feature of OqxB, play a crucial role in binding substrate molecules. Our point mutation studies of R157A also confirmed that this residue is crucial in Ciprofloxacin efflux.

With all these evidences, we feel that our DDM bound OqxB crystal structure facilitates the understanding of this important RND efflux pump and can be further utilized to delineate antibiotic/substrate binding.

If it is a substrate then can the authors explain why the protein is still in a symmetrical formation? I understood that RND proteins are in a symmetric "resting" state in the absence of substrate. However, when bound to a ligand they then separate into the different functional protomers. Thus, it would be good if authors could explain functional or physiological relevance of a symmetric homotrimer of OqxB with all monomers in binding conformation with two bound DDM molecules in each monomer; or if the occurrence of this structure is because of non-physiological conditions of crystallization. If the latter then could the authors provide MD simulation evidence, as they have shown the same thing for AcrB, that OqxB can exist as a functionally relevant asymmetric trimer.

It's a valid point and we have attempted to answer by performing aMD (accelerated MD simulations) without the DDM molecules in OqxB trimer. We have performed two important analyses to understand whether OqxB exists only in BBB/TTT conformation or shifts to other conformations like access/loose or extrusion/open. First, we have tracked cleft's opening and closing between the PC1 and PC2 domains, most of the available RND pump structures were used for distance calculations as well as MD simulation trajectories. We have also tracked helical nature of TM8 to demonstrate that conformational transitions from BBB/TTT of OqxB structure to EEE/OOO in the absence of ligands. We speculate that the pump will be in resting condition of EEE/OOO symmetric conformation and will transit from E/O to B/T conformation in the presence of substrates either through transient access/loose (A/L) conformation or directly.

Materials and Methods-

Why is an efflux inhibitor added- are results described using this? If DMSO is used as a solvent for some antimicrobial compounds what it really also used as a diluent for the 2-fold serial dilutions? Why was the MIC considered as 80% growth? This seems to be an arbitrary value. Why not a standard method?

The efflux inhibitor Phe-Arg β -naphthylamide (PA β N) was used for data in the supplementary Tables S4 and S5. It is used to demonstrate the low/lack of ciprofloxacin efflux by AcrB. The rest of the MIC experiments are without PA β N.

All the antimicrobial compounds were solubilised in DMSO (a standard solvent) and was used at a final concentration of 2 % for all tests, including the 2-fold dilutions. This ensured that there would be no inconsistency based on altered solubility and that all tests are done under identical conditions.

The MIC value was determined following a standard method as per the CLSI (2018) guidelines. MIC was determined as the concentration at which no bacterial growth was observed as ascertained by absorbance (A600nm) with a spectrophotometer. In case of the wild-type strain and clinical isolates, 90% growth inhibition was used for MIC determination. In the recombinant strains, the growth is compromised partially due to the presence of a plasmid and gene over-expression. Here, an 80% growth inhibition

results in the reduction of the test OD values to that of the media control levels and this is used for computation of MIC.

MD methodology -This sentence was copied directly from the website "The PPM server calculates rotational and translational positions of transmembrane and peripheral proteins in membranes using their 3D structure (PDB coordinate file) as input."

Is such detail required? "This server generates the input files required for various kinds of MD simulations, including bilayer embedded protein complex simulations."

We have removed the specified sentences as well as other parts which are not necessary.

Results

What is the AcrBperi structure?

We have added an explanation for AcrBperi as AcrB without the transmembrane portion. Thanks for the alert.

"To comprehend the binding mode of fluoroquinolones, computational molecular docking approach was utilized. Residues around DDM1 molecule were considered as binding pocket residues and docking grid was generated for docking simulations." Does this then mean that DDM2 is an artefact?

The sugar rings of DDM1 molecules reside in the distal binding pocket, whereas DDM2 occupies a probable proximal pocket. This we have shown in Figure S21 while comparing Erythromycin binding pocket. Based on the literature, it is well established that small molecular weight molecules bind at the distal binding pocket. In our observation, the DDM1 molecule binds at the distal binding pocket. This is the reason we have considered DDM1 molecule to generated docking grid.

"The asymmetric unit consists of six OqxB monomers/protomers and each monomer composed of 1040 residues." Really? This is contrasting with the authors' explanation of the OqxB structure (for example in the first paragraph of discussion, where they say it is symmetric structure). The information they have provided indicates that it is indeed symmetric structure.

Thank you for asking this. This is correct. But, it was not reader-friendly. The crystals belong to space group P1 form, and two trimers (=six monomers) are included in an asymmetric unit. So, there is no crystallographic symmetry between the structures, and they can be observed independently.

When the structures of each of the two trimers (and six monomers) were compared, all they were found to have similar structures as the "B (or T)" confirmation (RMDS < 1Å). So we should write as "non-crystallographic three-fold symmetry".

We corrected all the descriptions related to non-crystallographic symmetry.

The authors show that OqxB is functional in E. coli with additive efflux capacity for fluoroquinolone antibiotics in the WT and partnering ability with periplasmic adaptor and outer membrane channel components of the AcrAB-TolC system. However, the explanation and microbiology used is very hard to follow and needs to be rewritten. As it currently is Tables 1 and 2 do not easily stand on their own. This is primarily due to the nomenclature of deletion strains that is vague As an example:

C43(DE3) (D), C43(DE3) (D) *acrB*_OE, and C43(DE3) (D) *oqxB*_OE. Understanding of these strains is difficult with this nomenclature.

pEcoAB – which protein is the histag on?

The strain and the plasmid nomenclature have been changed and updated in the relevant tables.

Table 1 represents the antibiotic susceptibility done by overexpressing AcrA and B in the BW25113 wild-type background (reasons provided in the justification against the next comment). Table 2 represents the complementation done in the strain C43(DE3) Δ *acrB* (*acrB* deletion background). Both sets of data indicate that the heterologous expression of OqxB can complement the function of AcrB to efflux compounds. Our data suggest that the efflux of fluoroquinolones by OqxB is significant as compared to that by AcrB.

The strains and the plasmids are described in the Supplementary information. The strain details are added to the footnotes of Table 1 & Table 2 for the readers convenience.

***pEcoAB*, *pOqxAB* and His-Tag:**

None of the proteins carry a His Tag in the complementation assay where two genes are involved. For this assay, the construct was made using *acrAB* genes with a constitutive promoter and a *pTrc99a* vector backbone (NdeI-HindIII fragment, the *Trc* promoter also gets removed). The construct thus formed does not contain a His-Tag for any of the proteins. This information has been corrected and updated accordingly in Table S10 of the supplementary information.

Cloning Strategy: Only the 2.4kb digested vector backbone from pTrc99a carrying the Amp promoter-marker and pBR322-origin were used for cloning of double complementation system.

What is the rationale for complementing the wildtype *E. coli* strain bearing a natural chromosomal mediated *acrAB-tolC* system with a recombinant *acrAB* allele, and importantly what is the authors justification that this complementation has resulted in reduction of resistance to some of the AcrB substrates for example LZD, Cipro and Levo?

In *E. coli*, the expression of the *acrAB* operon is complex and is regulated by three activators, MarA, SoxS and Rob, and the repressor-AcrR. The objective here was to overcome the possible regulatory effects on the expression of *acrAB*, and ensure that their availability does not form a limiting factor. Therefore, the AcrAB was expressed with a constitutive promoter on a plasmid in wildtype *E. coli*. This was used to analyse its role in the efflux of various antibiotics, including the fluoroquinolones.

The 2-fold shift in MIC values for LZD, CIPRO and LEVO is a one well shift in the assay plates (based on OD of the culture and the cut-off considered). These observations are considered within the experimental variations of MIC experiments and are not significant differences.

Tables S4 and S5 need footnotes explaining the shading and underlining. Table S6 is not formatted correctly. What are the headers for the columns/rows?

Table S11 and S12 (re-numbered); the shading and the underlining were done to highlight the restriction sites for the individual primers. Elsewhere too, the formatting has been removed from the tables.

Figure S3 what are panels C and D?

Thanks for the alert; we have added the missing figure captions.

"This highly charged propensity of OqxB hydrophilic pocket is another remarkable difference that is not observed in any other RND efflux pumps (Supplementary Fig. S7)." Has the correct figure been referred to here?

Corrected the figure citation... now its Supplementary Fig. S7 (due to addition of Figure S6).

Overall, the English needs to be corrected. The article (the or a) is missing in multiple sentences. Italics needs to be checked when talking about genes and IS elements.

We have revised the manuscript by correcting the language to improve readability. Also, we have corrected the gene names as suggested.

REVIEWER COMMENTS

Reviewer #1 (Remarks to the Author):

I appreciate the efforts made by the authors to improve the manuscript. However, I have still some concerns that need to be addressed to support the publication of this work.

1. There are still no exhaustive details regarding the computational part. For instance, on page 9, can the authors specify at which temperature were the equilibration runs performed? Were the simulations performed at constant volume or constant pressure? Were the restraints reduced linearly or in steps? Which was the value of the average potential used for aMD simulations? Please, add detailed protocol and insert values where needed (see next comments regarding aMD simulations). As for experiments, also simulations must be reproducible.

2. ll. 472-3. Please amend the sentence to specify that the results refer to the strains considered in this work: "This data unequivocally indicated that ciprofloxacin has minimal efflux liability through the efflux pump AcrB"  "This data unequivocally indicated that ciprofloxacin has minimal efflux liability through the efflux pump AcrB in these strains".

3. Section "Probable resting-state determination by MD simulations". I strongly suggest the authors remove this section. aMD is a slow-converging technique, for which several microseconds are generally needed to achieve convergence on systems much smaller than RND transporters. The movements seen by the authors could well be oscillations around a conformation that is similar to the initial one. Additional and longer simulations are needed to support the authors' statement such as the one in the discussion ("MD simulations of OqxB in the absence of bound DDM molecules led to the conformational transition from BBB/TTT to EEE/OOO trimer.")

Finally, despite the author's claim that they improved the English, I still find their style redundant and not fluent.

In addition, the meaning of several sentences remains pretty unclear.

Thus, I reiterate my suggestion to improving the style (as this will also improve the impact).

Reviewer #2 (Remarks to the Author):

The authors have addressed the points raised by the referees including the ones by this referee. One of the points was the substitution of R157 and E50, where the authors made via site-directed mutagenesis the R157A variant, which confers fluoroquinolone susceptibility compared to wild type. The E50 residue was not addressed, but it is not clear whether the mutagenesis was not considered (and why not) or that maybe substitution did not cause an effect.

The authors also addressed the expression levels of *acrB*/*oqxB* by measuring the mRNA levels via qPCR. The suggestion was to measure AcrB/OqxB protein levels, but the MIC data were obtained by expression of non-tagged versions of these proteins, so mRNA level determination was conducted. The *oqxB* and *acrB* genes are expressed at different levels (Fig. S6). Do the authors think that the difference of *oqxB* expression compared to AcrB expression is causing the differences in the fluoroquinolone (and linezolid or novobiocin) susceptibilities seen in Table 1?

Reviewer #3 (Remarks to the Author):

The manuscript on the OqxB RND drug efflux protein from *Klebsiella pneumoniae* has been fully revised and modified. These changes have aided in the readability of the study and the extra data has either confirmed or extended the previous findings. There are still a few items that need

clarification/modification, some of which are specified below.

Line 238. Please state how many replicates of MIC analyses were performed. Are the MIC values presented an average or the representative of multiple replicates?

There are multiple instances where italics are either missing (gene and organism names) or should be removed. See lines 446-452 for multiple examples and Table S9. In Tables 1 and 2 the genes names need italicizing.

Line 451- change "&" to "and".

In some cases the English still needs attention especially in regards to verb tense, eg line 460 "reduce" should be "reduced".

Line 447 states that "MIC experiments confirmed its role in the extrusion of other common antibiotics with high efflux liability like linezolid, novobiocin and rifampicin (Table 2)."

This claim is not supported by the data provided in Table 2 for rifampicin.

Table 2 - Rifampin (Rifampicin) is a known substrate of AcrB as described in multiple references, eg Ma 1993, Molecular Cloning and Characterization of *acrA* and *acrE* genes of *Escherichia coli*; and Nikaido 1996, Multidrug Efflux Pumps of Gram-Negative Bacteria. What is the authors justification for not seeing this difference reflected in the delta AcrB variant of *E. coli* C43(DE3) for rifampicin as there is no difference in the MIC values for this delta *acrB* strain complemented with *oqxB*.

Reviewer #1 (Remarks to the Author):

I appreciate the efforts made by the authors to improve the manuscript. However, I have still some concerns that need to be addressed to support the publication of this work.

1. There are still no exhaustive details regarding the computational part. For instance, on page 9, can the authors specify at which temperature were the equilibration runs performed? Were the simulations performed at constant volume or constant pressure? Were the restraints reduced linearly or in steps? Which was the value of the average potential used for aMD simulations? Please, add detailed protocol and insert values where needed (see next comments regarding aMD simulations). As for experiments, also simulations must be reproducible.

Thank you for the alert about technical details that were missing, we have now included the cMD simulations set-up details as well as aMD parameters which we used for Apo simulations.

2. II. 472-3. Please amend the sentence to specify that the results refer to the strains considered in this work: “This data unequivocally indicated that ciprofloxacin has minimal efflux liability through the efflux pump AcrB”  “This data unequivocally indicated that ciprofloxacin has minimal efflux liability through the efflux pump AcrB in these strains”.

Thank you for the suggestion, we have now updated the text as advised.

3. Section “Probable resting-state determination by MD simulations”. I strongly suggest the authors remove this section. aMD is a slow-converging technique, for which several microseconds are generally needed to achieve convergence on systems much smaller than RND transporters. The movements seen by the authors could well be oscillations around a conformation that is similar to the initial one. Additional and longer simulations are needed to support the authors’ statement such as the one in the discussion (“MD simulations of OqxB in the absence of bound DDM molecules led to the conformational transition from BBB/TTT to EEE/OOO trimer.”)

We appreciate the reviewers concern and to tone down our claims we have moved “Probable resting-state determination by MD simulations” part to supporting information. We agree with the reviewer that aMD simulations takes longer time scales to reach the complete conformational transitions from Binding to Extrusion and especially in our case three binding protomers to three extrusion protomers. As mentioned in the manuscript, lacking other protomer conformations hindered the tMD simulations which is a better approach as these are large complex systems.

We have analyzed aMD simulations outcome by two methods i.e., 1) Distance measurement between PC1 and PC2 subdomain residues 2) extended helical nature of TM8. These analyses demonstrated that the B chain of the BBB trimer probably shifted to extrusion conformation as the TM8 helix extension observed around 50ns simulation time and continued to be in same extended conformation during the remaining time of the simulation.

To minimize the impact of our claims we have dropped the sentence “MD simulations of OqxB in the absence of bound DDM molecules led to the conformational transition from BBB/TTT to EEE/OOO trimer.” from the discussion section.

Finally, despite the author's claim that they improved the English, I still find their style redundant and not fluent.

In addition, the meaning of several sentences remains pretty unclear.

Thus, I reiterate my suggestion to improving the style (as this will also improve the impact).

As advised, the manuscript now proof-read and edited by Dr. Ed Griffen (MedChemica Ltd., England)

Reviewer #2 (Remarks to the Author):

The authors have addressed the points raised by the referees including the ones by this referee. One of the points was the substitution of R157 and E50, where the authors made via site-directed mutagenesis the R157A variant, which confers fluoroquinolone susceptibility compared to wild type. The E50 residue was not addressed, but it is not clear whether the mutagenesis was not considered (and why not) or that maybe substitution did not cause an effect.

Due to the paucity of time, we have only presented the data for R157A. E50A, along with other mutants and its efflux efficiency and selectivity are being currently evaluated and will be presented in a subsequent publication.

The authors also addressed the expression levels of *acrB*/*oqxB* by measuring the mRNA levels via qPCR. The suggestion was to measure *AcrB*/*OqxB* protein levels, but the MIC data were obtained by expression of non-tagged versions of these proteins, so mRNA level determination was conducted. The *oqxB* and *acrB* genes are expressed at different levels (Fig. S6). Do the authors think that the difference of *oqxB* expression compared to *AcrB* expression is causing the differences in the fluoroquinolone (and linezolid or novobiocin) susceptibilities seen in Table 1?

The data (Table 1 and Fig. S6) clearly indicates that in the Wild Type *E. coli*, a homologous over expression of *AcrB* by 5- fold does not shift the MIC of Ciprofloxacin. In contrast, a heterologous overexpression of *OqxB* by 3-fold (in comparison to the *AcrB* Over-expression), causes an upshift of MIC for Ciprofloxacin, and other Fluoroquinolones by 8-16 folds. It should also be noted that the overexpression of *AcrB* or *OqxB* do not indicate a proportional increase of functional efflux complex. The association of the TolC protein, (which is at normal basal levels in the current scenario) to either *AcrAB* or *OqxAB* is stoichiometrically constrained. This indicates that the over expression of individual pump proteins does not translates linearly to the assembly of a functional tripartite efflux pump, as indicated by the fold change of the transcripts.

Reviewer #3 (Remarks to the Author):

The manuscript on the *OqxB* RND drug efflux protein from *Klebsiella pneumonia* has been fully revised and modified. These changes have aided in the readability of the study and the extra data has either confirmed or extended the previous findings. There are still a few items that need clarification/modification, some of which are specified below.

Line 238. Please state how many replicates of MIC analyses were performed. Are the MIC values presented an average or the representative of multiple replicates?

Thank you for the alert, the experiments were performed in triplicates and the reported MIC values are representative of the data obtained from those set-ups. The same has been updated in the Materials and Methods section of the manuscript.

There are multiple instances where italics are either missing (gene and organism names) or should be removed. See lines 446-452 for multiple examples and Table S9. In Tables I and 2 the genes names need italicizing.

Thank you for the suggestion, the corrections have been incorporated, as advised.

Line 451- change “&” to “and”.

The corrections have been incorporated, as suggested.

In some cases the English still needs attention especially in regards to verb tense, eg line 460 “reduce” should be “reduced”.

As advised, the manuscript now proof-read and edited by Dr. Ed Griffen (MedChemica Ltd., England)

Line 447 states that “MIC experiments confirmed its role in the extrusion of other common antibiotics with high efflux liability like linezolid, novobiocin and rifampicin (Table 2).”

The corrections for reporting Rifampicin as efflux liable compound made in the manuscript. Rifampicin has been considered as a negative control (with minimal efflux liability) in our studies.

This claim is not supported by the data provided in Table 2 for rifampicin.

Table 2 - Rifampin (Rifampicin) is a known substrate of AcrB as described in multiple references, eg Ma 1993, Molecular Cloning and Characterization of *acrA* and *acrE* genes of *Escherichia coli*; and Nikaido 1996, Multidrug Efflux Pumps of Gram-Negative Bacteria. What is the authors justification for not seeing this difference reflected in the delta AcrB variant of *E. coli* C43(DE3) for rifampicin as there is no difference in the MIC values for this delta *acrB* strain complemented with *oqxB*.

- **Rifampicin has been used as one of antibiotics for negative control (Zero to minimal efflux liability in our MIC assays and we have not observed any shifts in between the wild type and the delta AcrB (*ΔacrB*) variants.**
- **According to the reviewer’s suggestion, the Minireview (Multidrug Efflux Pumps of Gram-Negative Bacteria by Nikaido,1996) although mentions Rifampicin as one of the substrates for *E. coli* AcrA-AcrB-TolC pumps, there is no information available for the MIC values for the same. Further referring to the cross-references mentioned in the aforesaid Minireview and the reviewer’s suggestion, the paper entitled “Molecular Cloning and Characterization of *acrA* and *acrE* Genes of *Escherichia coli*” (J Bacteriol. 1993 Oct;175(19):6299-313. doi: 10.1128/jb.175.19.6299-6313.1993), mentions the drug susceptibility in *E. coli* W4573 (F- K-12 lac ara mal xyl mtl gal rpsL) and its isogenic *acrA* mutant derivative N43 (W4573 *acrA1*). The reported MIC values for Rifampicin for the *E. coli* W4573 was 16 µg/ml, the corresponding value for the *acrA* mutant derivative was reported to be 8 µg/ml. A 2-fold shift in the MIC values is considered within the experimental errors of MIC assay experiments. This observation, also supports and confirms our findings that Rifampicin has a minimal efflux liability via AcrA-AcrB-TolC complex in *E. coli*.**

REVIEWERS' COMMENTS

Reviewer #1 (Remarks to the Author):

The authors have addressed the points I made, thus I suggest publication of their work.